# Vicarious reward unblocks associative learning about novel cues in male rats

Sander van Gurp[1]*, Jochen Hoog[1,2], Tobias Kalenscher[2], Marijn van Wingerden[1,3]

[1]Social Rodent Lab, Institute of Experimental Psychology, Heinrich-Heine-University, Düsseldorf, Germany; [2]Comparative Psychology, Institute of Experimental Psychology, Heinrich-Heine-University, Düsseldorf, Germany; [3]Department of Cognitive Science and Artificial Intelligence, Tilburg School of Humanities and Digital Sciences, Tilburg University, Tilburg, Netherlands

**Abstract** Many species, including rats, are sensitive to social signals and their valuation is important in social learning. Here we introduce a task that investigates if mutual reward delivery in male rats can drive associative learning. We found that when actor rats have fully learned a stimulus-self-reward association, adding a cue that predicted additional reward to a partner unblocked associative learning about this cue. By contrast, additional cues that did not predict partner reward remained blocked from acquiring positive associative value. Importantly, this social unblocking effect was still present when controlling for secondary reinforcement but absent when social information exchange was impeded, when mutual reward outcomes were disadvantageously unequal to the actor or when the added cue predicted reward delivery to an empty chamber. Taken together, these results suggest that mutual rewards can drive associative learning in rats and is dependent on vicariously experienced social and food-related cues.

*For correspondence:
sander.gurp@hhu.de

**Competing interests:** The authors declare that no competing interests exist.

## Introduction

Humans and other animals have developed a capacity for mutual cooperative behavior (*Nowak, 2006*; *Rand and Nowak, 2013*; *Rilling et al., 2002*; *Suchak et al., 2014*), a preference for prosocial outcomes to familiar partners (*Hernandez-Lallement et al., 2015*; *Horner et al., 2011*; *Márquez et al., 2015*) and helping behavior toward others in need (*Ben-Ami Bartal et al., 2011*; *Fehr and Rockenbach, 2004*). These behaviors are sometimes costly, prompting questions why actor rats engage in them (*de Waal and Suchak, 2010*; *Hamilton, 1963*; *Stevens et al., 2005*; *Trivers, 1971*). Some researchers have focused on putative future reciprocation (*Taborsky et al., 2016*) as a potential driver, while others have highlighted that acting generously could generate self-reward internally (*Harbaugh et al., 2007*; *Park et al., 2017*) or through positive social reward signals such as friendly faces in humans (*Spreckelmeyer et al., 2009*). Indeed, the capacity to identify positive, rewarding outcomes delivered to others is a fundamental aspect of social observational learning (*Zentall, 2012*). Underlying some of these suggestions is the assumption that rewarding outcomes to a social partner could also represent value to oneself and thus drive a proximate reward/learning mechanism (*Hernandez-Lallement et al., 2016*; *Ruff and Fehr, 2014*). Using this logic, animals, including humans, choose prosocial outcomes, cooperate or act altruistically because these actions result in vicarious reward, experienced through sensitivity to the behavioral and/or affective state of the partner (*de Waal and Preston, 2017*; *Prochazkova and Kret, 2017*), in addition to putative anticipated future reciprocal reward (*Taborsky et al., 2016*). One important aspect of social learning is identifying the features of the environment that predict (vicariously) rewarding outcomes, and learning the (instrumental) action sequence, appropriate to the context, for acquiring these vicariously rewarding outcomes. There is evidence that cues that predict social reward can become valuable as humans learn to respond faster to stimuli that become associated with positive social

reinforcement (*Jones et al., 2011*) and monkeys preferred stimuli that predicted a reward delivery to a conspecific more than the stimuli that predicted no reward delivery (*Chang et al., 2011*). In rats, it was found that observing another rat being rewarded is (vicariously) rewarding by itself as it is accompanied by 50 kHz vocalizations, indicative of a positive appetitive state (*Burgdorf et al., 2011*; *Panksepp, 2007*), and dopamine release in the NAcc of the observer rat (*Kashtelyan et al., 2014*). Indeed, playback of 50 kHz leads to both an approach response (*Wöhr and Schwarting, 2007*) and results in dopamine release in the Nucleus Accumbens NAcc (*Willuhn et al., 2014*). We, therefore, hypothesized that vicarious reward, associated with rewards delivered to others, could also reinforce Pavlovian associative learning about novel cues, as has been found in the appetitive domain (*Berridge, 2012*; *Schultz, 2016*). To investigate our hypothesis, we use a well-established behavioral paradigm in associative learning called blocking. *Kamin, 1969* found, in simple stimulus-outcome association tasks, that if new stimuli are added to a stimulus that already fully predicts a reward, associative learning about those additional stimuli will be blocked. Reinforcement learning about additional stimuli can become unblocked, however, by an increase in reward value or a change in reward identity contingent on the presentation of the new stimuli. This change in value is then thought to be associated with these new stimuli and thus alters their incentive value (*Holland, 1984*). We hypothesize that rewarding social outcomes, such as sugar pellet deliveries to a partner rat, will also be capable to unblock learning about novel stimuli added in compound, indicative of an increased, partially vicarious value of mutual rewards relative to own-rewards. We tested this hypothesis by adopting a task from *McDannald et al., 2011* where unblocking is operationalized by adding additional pellet deliveries conditional on a second cue presented in compound with a learned cue that already fully predicted reward. We modified this task in such a way that the second cue is now followed by a food reward delivery to a partner rat, rather than increasing one's reward. In addition, a third control cue added in compound to the learned cue (on different trials) was not followed by food reward delivery to a partner rat. Concretely, we thus hypothesized that associative learning about the second stimulus would become unblocked through a vicarious experience of the partner reward exclusively during mutual reward outcomes. By contrast, the third cue should remain blocked from acquiring associative value due to the absence of a reward outcome for the partner. We indeed found, when tested in extinction, that the unblocked cue had acquired more associative value, as indexed by conditioned responding at the food trough, in comparison to the blocked cue. Importantly, this effect was still present when controlling for potential effect of secondary reinforcement associated with increased pellet deliveries. Crucially, this difference was absent (1) when social information exchange was impeded, (2) when the partner rat was absent during mutual reward delivery, and (3) when the unblocking cue was associated solely with partner reward but not actor reward, presenting a disadvantageous unequal reward distribution to the actor rat.

We thus conclude that mutual, equal reward delivery can trigger a positive vicarious reward experience that supports unblocking of associative learning about novel cues. This opens up possibilities to investigate behavioral aspects of the social-value driven reinforcement learning and its associated neural basis, processes that might be disturbed in psychiatric disorders marked by impaired reinforcement learning and/or social behavior such as autism (*Kohls et al., 2012*) and schizophrenia (*Fulford et al., 2018*).

## Results

All groups of actor and partner rats were initially trained separately on a Pavlovian discrimination problem. Subsequently, the rats went through a social learning phase were actor rats could learn to associate additional compounded cues with different reward outcomes delivered to the partner rat (social unblocking). Finally, we tested the associative strength of all cues, each presented in isolation, in a probe phase without a reward. In the inserted wall control experiment, we impeded the exchange of visual information by implementing an opaque wall and in the no partner present control experiment, we implemented the social learning phase without a partner rat present. Finally, in the unequal outcomes control experiment, we implemented the social learning phase with unequal, disadvantageous reward outcomes (see *Figure 1* for the experiment timeline and *Figure 1—figure supplement 1* for the group overview). We illustrate the actor rats' conditioned responses with the time spent in the food cup, the food cup rate, and their latency to entry as dependent variables. We subdivide the result section into two parts. We demonstrate that cues that predicts no additional

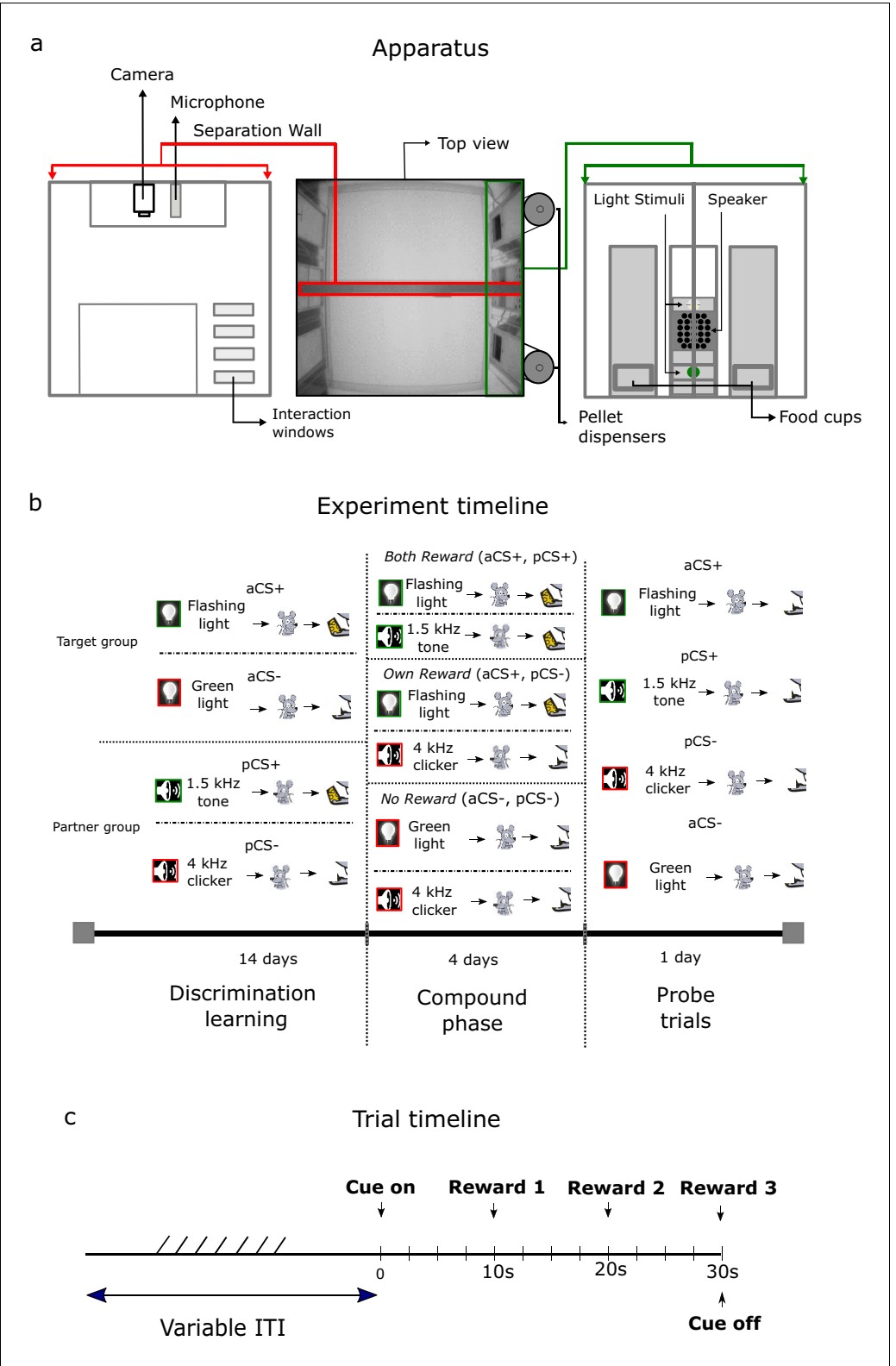

**Figure 1.** Behavioral apparatus, experimental timeline, and trial timeline. (a) The PhenoTyper consisting of lower and upper compartment in which behavioral training took place is displayed in the middle. On the left, the custom made separation wall is shown with interaction windows, camera, and microphone. On the right, the right side of the PhenoTyper is displayed with the used operant devices and in both sides of the box the food cup. (b) An example experimental time line is displayed. Actor rats learn to discriminate two visual cues in the upper

*Figure 1 continued on next page*

*Figure 1 continued*

compartment while at a different time partner rats learned to discriminate two auditory cues in the lower compartment. In the compound phase actor and partner rat are either both rewarded (BR, aCS+/pCS+), actor rat is rewarded while the partner is not rewarded (OR, aCS+/pCS-) or both actor and partner rat are not rewarded (NR, aCS/pCS-). In the probe trials, all learned cues are presented to the actor and at a different time to the partner rat without reward. (c) Here a timeline is shown with the different components that make up a single trial throughout the discrimination learning, compound phase, and probe trials.

The online version of this article includes the following figure supplement(s) for figure 1:

**Figure supplement 1.** Group assignments and pellet dispenser configuration over different experiment subgroups.

**Figure supplement 2.** Cue overview for each subgroup.

---

reward for the partner are blocked in both the experimental group and all control groups. We then show that, generally, we find vicarious unblocking for food cup occupancy in experimental group 1 (combined social-appetitive and social-only subgroups) but not in control group 1 (combined inserted wall and no partner present subgroups) and control group 2 (unequal outcomes). We furthermore examine the pattern of unblocking over time and in addition, we investigate potential identity unblocking by looking at the food cup rate. The second part of the results section presents several control experiments that show that the vicarious unblocking response is still present when controlling for secondary reinforcement of pellet dispenser sounds but not when a wall was placed between partners to prevent social information exchange (control group 1, inserted wall subgroup). Likewise, unblocking was diminished when there was no partner present during the social learning phase (control group 1, no partner present subgroup).

Discrimination learning. Actor rats (N = 20) were trained on a visual or auditory discrimination task with counterbalanced exemplars as aCS+ and aCS- stimuli (see *Figure 1—figure supplement 2*). All actor rats developed a conditioned response to their own aCS+ (see example trial in *Video 1*), resulting in an increase with learning in time spent in the food trough on aCS+ trials in anticipation of reward, independent of cue modality (see below). Concurrently, they learned to expect no reward during aCS- presentations, as witnessed by a steady decrease in time spent in the food trough on aCS- trials (*Figure 2a,c,e*). A paired sample *t*-test examining the mean responding over the last 4 days of conditioning was performed. We found a significant difference in time spent in the food trough between the aCS+ and aCS- of the experimental group (M = 58.76, SD = 12.86; M = 21.19, SD = 13,21; $t(19)$ = 12.116, p<0.001), control group 1 (M = 54.649, SD = 14,604; M = 15.61, SD = 7.86; $t(19)$ = 13.472, p<0.001) and control group 2 (M = 53.82, SD = 18.06; M = 17.66, SD = 9.02; $t(19)$ = 7.57, p<0.001). We performed a two-way ANOVA to assess whether discrimination ability was similar in the experimental conditions and for the different stimulus types (auditory or visual) using the difference scores (aCS+/aCS-) on the last 4 days of training. There was no significant difference between groups ($F_{(2, 46)}$=0.141, p=0.869), no difference between auditory and visual discrimination learning ($F_{(1, 46)}$=0.076, p=0.785) and finally no interaction between experiment and stimulus type ($F_{(2, 46)}$=0.297, p=0.745).

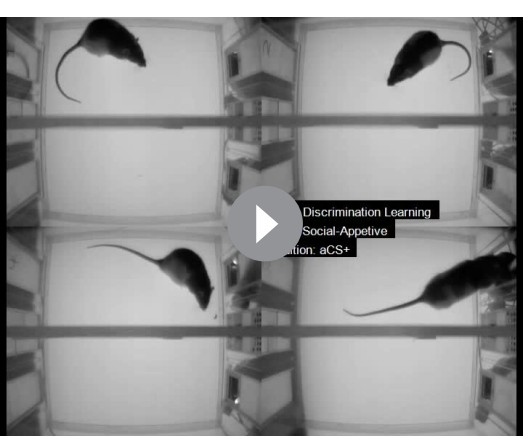

**Video 1.** Conditioned response of the actor rats during the presentation of the aCS+ on day 14 of discrimination learning. This video shows the conditioned response of the actor rats of the social-appetitive subgroup on trial 1 day 14. Shown here are the 5 s before cue onset and the first 10 s after cue onset.

https://elifesciences.org/articles/60755#video1

## Social learning

In this phase, rats were trained together. The aCS+/aCS- of the actor and pCS+/pCS- of the partner were combined in three compound combinations with the following reward outcomes:

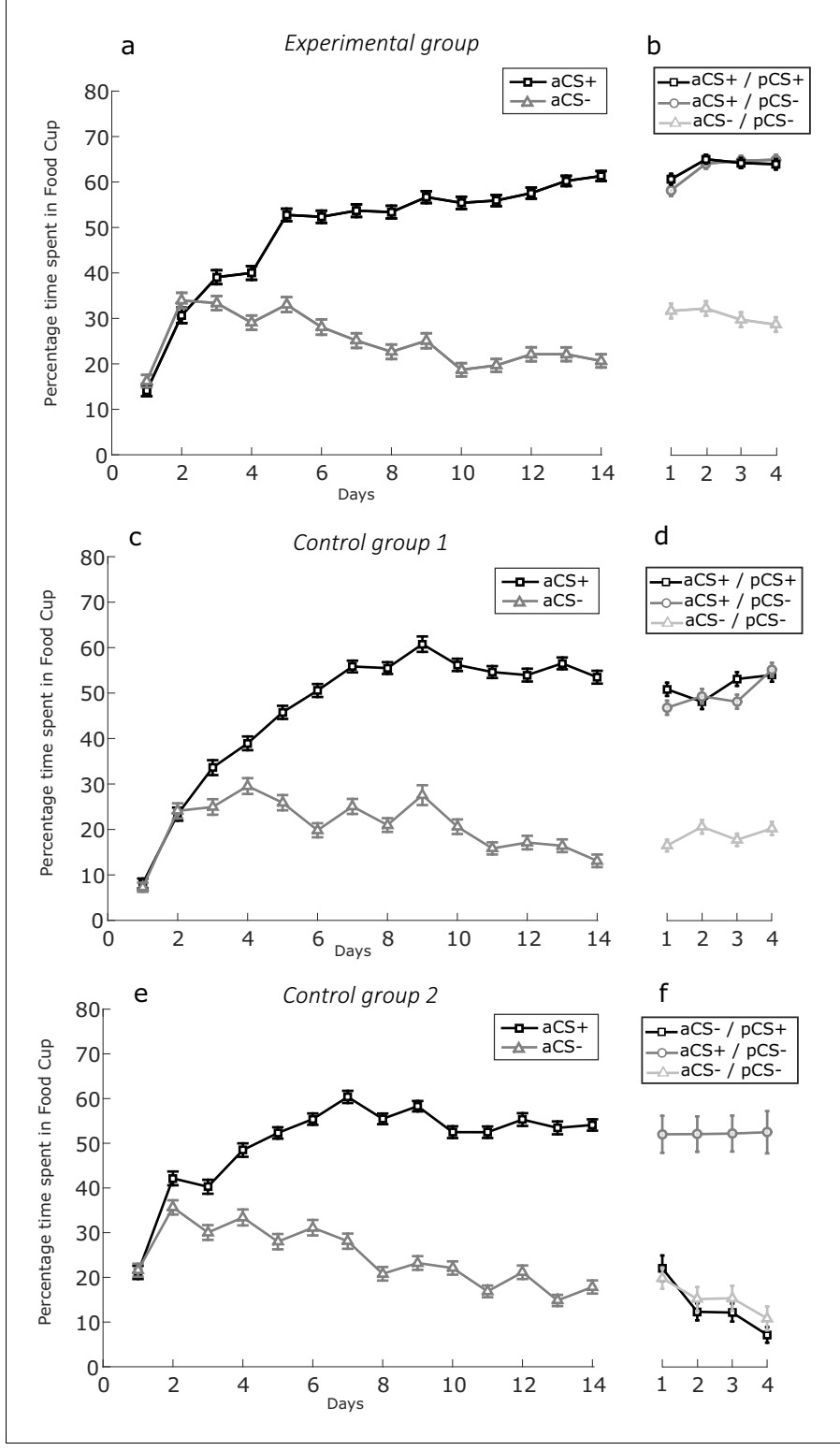

**Figure 2.** Conditioning per experimental phase. Experimental group (combined social-appetitive and social-only subgroups). (a) Percentage of time spent in food cup for discrimination learning between aCS+ and aCS- over days. (b) Percentage of time spent in food cup for the compounds BR (aCS+, pCS+), OR (aCS+, pCS-), and NR (aCS-, pCS-) over days. Control group 1 (combined inserted wall and no partner present subgroups). (c) Percentage of time spent in food cup for discrimination learning between aCS+ and aCS- over days. (d) Percentage of time spent in food cup for the compounds BR (aCS+, pCS+), OR (aCS+, pCS-), and NR (aCS-, pCS-)
*Figure 2 continued on next page*

*Figure 2 continued*

over days. Control group 2 (unequal outcomes). (**e**) Percentage of time spent in food cup for discrimination learning between aCS+ and aCS- over days. (**f**) Percentage of time spent in food cup for the compounds PR (aCS-, pCS+), OR (aCS+, pCS-), and NR (aCS-, pCS-) over days. Error bars indicate SEM.

The online version of this article includes the following figure supplement(s) for figure 2:

**Figure supplement 1.** Effect of adding a novel cue on the aCS+.

---

both reward (BR: aCS+/pCS+), own reward (OR: aCS+/pCS-), and no reward (NR: aCS-/pCS-). In the main experiments, we chose to omit the 'partner reward' condition where the target rats would *not* receive reward, while the partner rats would (PR: aCS-/pCS+; but see unequal outcomes control), to avoid a potential reward/value conflict due to disadvantageous inequity aversion (*Fehr and Schmidt, 1999*), which has been reported in rats as well (*Oberliessen et al., 2016*). Rats' conditioned responses to these compound cues are shown (for an example trial see *Videos 2* and *3*) and a direct comparison of these responses to the original aCS+ and aCS- cues was made, both indexed by time spent in the food cup and the food cup rate (*Figure 2b,d,f* and *Figure 2—figure supplement 1*) In the subsequent analysis, only the behavior of the actor rats is reported. We applied a mixed repeated measures ANOVA design with the three compound trial types (BR [partner reward (PR) for control group 2], OR, and NR) and day 1-4 as within-subject factors, and with group (experimental vs control 1 vs control 2) as between-subject factor. The time spent in the food cup during the first 10 s after the cue onset was chosen as the dependent variable. We found a significant main effect of trial type ($F_{(1.568, 76.845)}$=161.520, p<0.001, $\eta_p^2$ = 0.767) and importantly found an interaction effect of experiment * trial type ($F_{(3.137, 76.845)}$=28.243, p<0.001, $\eta_p^2$ = 0.537), reflecting the difference in experiments that employed BR versus PR trials; and no effect of day ($F_{(2.223, 108.195)}$=0.017, p<0.001, $\eta_p^2$ = 0.997). Post-hoc comparison revealed that actors' responding to the BR cue did not differ significantly from the OR cue in experimental group 1 (mean difference = 0.490, SE = 2.367, p=1.00) and control group 1 (mean difference = 1.603, SE = 2.646, p=1.00), while in control group two responding is smaller in PR (partner reward) than OR trials (mean difference = −38.784, SE = 2.646, p<0.001). BR responding was furthermore significantly higher than NR in experimental group 1 (mean difference = 32.9840.784, SE = 2.848, p<0.001) and control group 1 (mean difference = 32.684, SE = 3.185, p<0.001), while in control group 2, PR responding was not significantly different from NR (mean difference = −1.833, SE = 3.185, p=1.00), arguing against social facilitation of conditioned responding as a social learning mechanism. Finally, OR responding was significantly higher than NR responding in all groups (experimental group 1: mean difference = 32.495, SE = 3.704, p<0.001; control group 1: mean difference = 31.081, SE = 4.141, p<0.001; control group 2: mean difference = 36.901, SE = 4.141, p<0.001). We furthermore assessed if the average compound phase food cup responses over 4 days changed in comparison to the last 4 days of discrimination learning due to the addition of pCS+ and pCS- cues. Next, we assessed if there were any between-group and within-condition differences in the compound phase food cup responses to BR, OR, and NR cues. We first ran a mixed repeated measures ANOVA analysis, with three difference scores (aCS+/aCS-, BR/NR, and OR/NR) as within-subject factors and group (experimental vs control 1 vs control 2) and stimulus type (auditory/visual) as between-subject factors (*Figure 2—figure supplement 1a,b and c*). We found a significant main effect

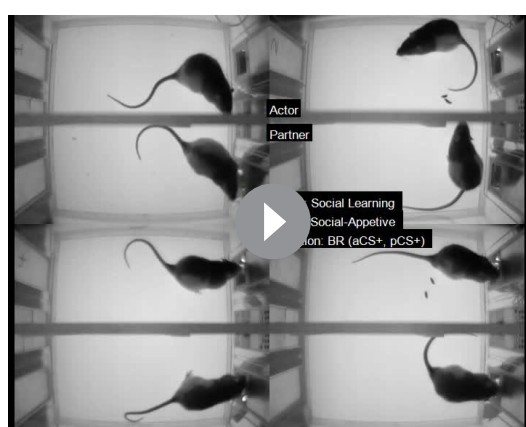

**Video 2.** Conditioned response of the actor and partner rats during the presentation of the BR (aCS+, pCS+) on day 4 of social learning. This video shows the conditioned response of both the actor (top compartment) and partner (bottom compartment) rats of the social-appetitive subgroup on trial 1 day 4. Shown here are the 5 s before cue onset and the first 20 s after cue onset.
https://elifesciences.org/articles/60755#video2

of trial type ($F_{(1.435, 66.017)}$=35.071, p<0.001, $\eta_p^2$ = 0.433), we furthermore found an interaction effect of trial type * group ($F_{(2.870, 66.017)}$=22.188, p<0.001, $\eta_p^2$ = 0.491), an interaction effect of trial type * stimulus type ($F_{(1.435, 66.017)}$=4.286, p=0.029, $\eta_p^2$ = 0.085) but no effect of experiment * trial type * stimulus type ($F_{(2.870, 66.017)}$=1.577, p=0.187, $\eta_p^2$ = 0.064). Post-hoc comparison for the trial type * stimulus type interaction found that rats, over all experiments, have a significantly smaller contrast score for visual than auditory cues in the OR/NR contrast (mean difference = −10.862, SE = 4.141, p<0.001), near significantly smaller in the BR/NR (mean difference = −6.409, SE = 3,432, p=0.068) but not smaller for the aCS+/aCS- (mean difference = +1.191, SE = 4.331, p=0.785). There are furthermore no differences within-experiments between aCS+/aCS- contrast scores and BR/NR or OR/NR contrast scores in all experimental groups, arguing against a putative effect on associative learning of the compound cues due to more vigorous responding in the compound phase. The only expected differences observed here is that the PR/NR contrast in control experiment 2 is smaller than the OR/NR (mean difference = 38.784, SE = 2.532, p<0.001) and aCS+/aCS- contrast (mean difference = 38.044, SE = 3.676, p<0.001), because of the altered reward contingencies. Finally, most importantly, we do not find any differences between contrasts between different experimental groups with the only exception again for the PR/NR contrast which is lower for the control group 2 (PR) compared experimental group 1 (BR) and control group 1 (BR). We find that the observed effect of stimulus type is mostly captured by a slight shift in conditioned responding to the NR compound in comparison to the aCS- responses during discrimination learning. When running a mixed repeated measures ANOVA design, with the aCS- and NR as within-subject factors and group (experimental vs control 1 vs control 2) and stimulus type (auditory/visual), we find a triple interaction effect (*Figure 2—figure supplement 1d,e and f*; F(2,46) = 5.247, p=0.009, $\eta_p^2$ = 0.186). It becomes clear that the rats show significantly more food cup responses to the visual cues in comparisons to the auditory cues in the NR (mean difference = 18.966, SE = 3.061, p<0.001) compared to the aCS- in experimental group 1 but not in control groups 1 and 2. Conditioned responses to visual cues are furthermore higher in NR over aCS- in experimental group 1 (mean difference = 25.247, SE = 4.412, p<0.001) and near significantly higher in control group 1 (mean difference = 9.00, SE = 4.697, p=0.062) but not in control group 2.

These results indicate that adding an additional cue predicting a BR or OR outcome does not change the conditioned response in comparison to NR during discrimination learning in the experimental and control groups. Importantly, no differences were observed between rewarded conditions indicating that partner presence does not influence food cup responses by itself. The only difference we notice is that adding a visual cue to an auditory cue leads to increased food cup response in the NR condition experimental group 1 and control group 1 but not in control group two compared to the aCS-. This could indicate a deficit in inhibitory action control of a learned auditory CS- because of partner presence, or reflect some difference in stimulus efficacy or asymmetrical processing interacting with social partner presence that cannot be entirely interpreted.

## Probing vicarious associative learning

In the probe trials, we aimed to show the effect of associative learning driven by self and vicarious reward. In an extinction setting, rats were individually exposed to the cues in isolation (i.e. one at a time), omitting reward. The learned associative value of each cue was indexed by the time spent in the food cup, the food cup rate, and the latency to entry over 10 extinction trials per cue (the presentation order of cues was intermixed). We show the percentage conditioned responding of the actor rats to the first 10 s of 10 presentations each of the aCS+ and aCS-, the pCS+ (unblocked) cue (example trial in *Video 4*) associated with an added reward to the partner (BR) and the pCS- (blocked) cue (example trial in *Video 5*) associated with no additional reward to the partner (OR) and no reward to self (NR) (*Figure 3a, b, c*) We binned responses in groups of two trials. Summary statistics for the *Figure 3* comparisons (F-stats, p-values, effect sizes), including the time spent in the food cup for the 30 s after cue onset (*Figure 3d,e and f*), can be found in *Figure 3—source data 1*.

## Vicarious reward unblocks associative learning

A two-factor repeated measures ANOVA with stimulus type and bin as factors and the time spent in the food trough as the dependent variable was performed for the experimental group (combined social-appetitive and social-only subgroups). To sum up, we found a significant probe phase main

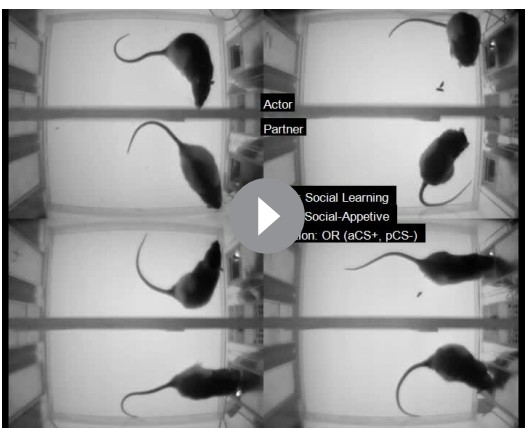

**Video 3.** Conditioned response of the actor and partner rats during the presentation of the OR (aCS+, pCS-) on day 4 of social learning. This video shows the conditioned response of both the actor (top compartment) and partner (bottom compartment) rats of the social-appetitive subgroup on trial 2 day 4. Shown here are the 5 s before cue onset and the first 20 s after cue onset.

https://elifesciences.org/articles/60755#video3

effect of cue type on time spent in food trough ($F_{(3, 57)}$=83.180, p<0.01, $\eta_p^2$ = 0.814). As expected, time spent in the food trough was higher for aCS+ than aCS- trials (mean difference = 37.322, SE = 2.958, p<0.001; Cohen's d = 1.92, *Figure 3a*). Critically, pairwise comparisons revealed that the actors spent more time in the food trough for pCS+ (unblocked) cues than for pCS- (blocked) cues (mean difference = 10.804, SE = 2.592, p=0.003, Cohen's d = 0.60; *Figure 3a*). Furthermore, we also found a significantly higher responding to the pCS+ cue compared to the aCS- cue (mean difference = 14.544, SE = 2.257, p<0.001, Cohen's d = 0.88) while responding to the pCS-cue was not significantly different from the aCS-cue (mean difference = 3.740, SE = 1.190, p=0.390, Cohen's d = 0.25), suggesting that the blocked cue is treated similarly to the aCS-, in line with learning theory. Additionally, we found a main effect of bin number on time spent in food trough ($F_{(4, 76)}$=18.678, p<0.01, $\eta_p^2$ = 0.496), reflecting the extinction process, and finally, we found an interaction between cue type and bin number on the time spent in food trough ($F_{(12,228)}$=2.930, p=0.001, $\eta_p^2$ = 0.137). Simple effects analysis revealed that the food cup response for the pCS+ was significantly higher than pCS-for bin 1 (mean difference = 17.200, SE = 5.461, p<0.032), bin 3 (mean difference = 19.360, SE = 0.007, p=0.007) but that this difference disappeared from bin 4 (mean difference = 4.830, SE = 7.076, p=1.00).

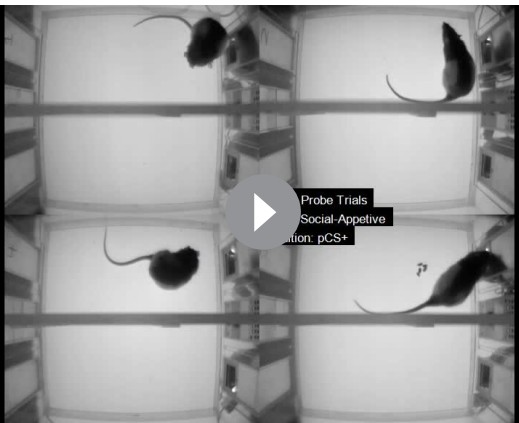

**Video 4.** Conditioned response of the actor rats during the presentation of the pCS+ on trial 1 of the probe trials. This video shows the conditioned response the actor rats of the social-appetitive subgroup on trial 1 of the pCS+ during the probe trials. Shown here are the 10 s before cue onset and the first 10 s after cue onset.

https://elifesciences.org/articles/60755#video4

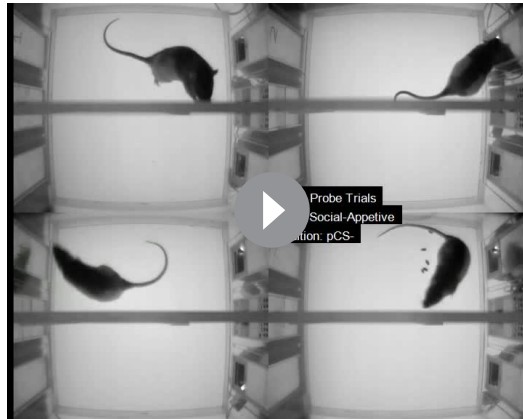

**Video 5.** Conditioned response of the actor rats during the presentation of the pCS- on trial 1 of the probe trials. This video shows the conditioned response the actor rats of the social-appetitive subgroup on trial 1 of the pCS- during the probe trials. Shown here are the 10 s before cue onset and the first 10 s after cue onset.

https://elifesciences.org/articles/60755#video5

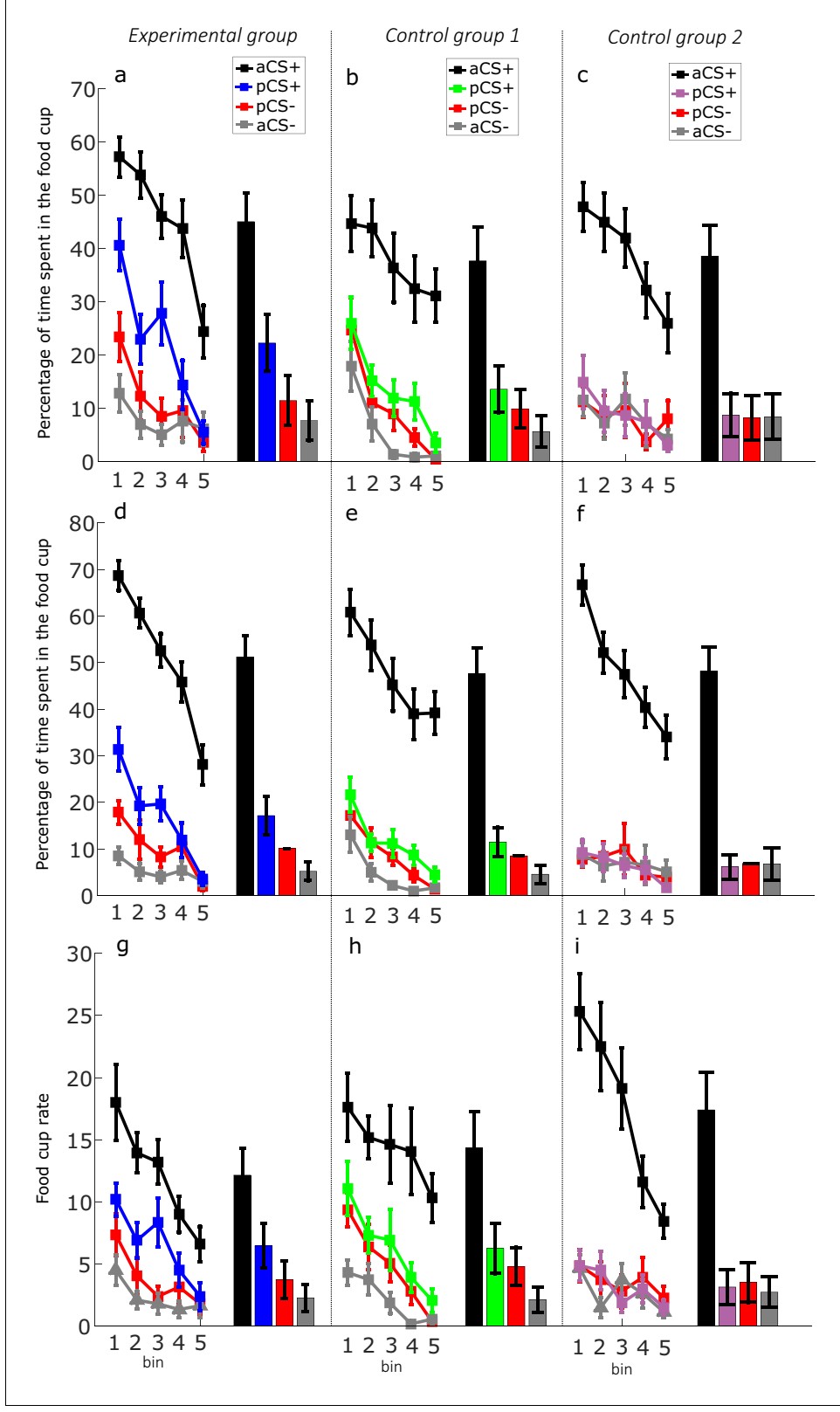

**Figure 3.** Food cup response during the probe trials. *Experimental group 1 (combined social-appetitive and social-only subgroups).* (a) Percentage of time spent in the food cup in the 10 s period after cue onset during extinction. (d) Percentage of time spent in the food cup in the 30 s period after cue onset (g) Food cup rate per minute in the 10 s period after cue onset. *Control group 1 (combined inserted wall and no partner present Figure 3 continued on next page*

*Figure 3 continued*

subgroups). (**b**) Percentage of time spent in the food cup in the 10 s period during extinction after cue onset. (**e**) Percentage of time spent in the food cup in the 30 s period after cue onset. (**h**) Food cup rate per minute in the 10 s period after cue onset. *Control group 2 (unequal outcomes).* (**c**) Percentage of time spent in the food cup in the 10 s period after cue onset during extinction. (**f**) Percentage of time spent in the food cup in the 30 s period after cue onset. (**i**) Food cup rate per minute in the 10 s period after cue onset. Extinction in all groups is depicted in five bins of two trials per bin. Bar plots indicate averaged time spent in the food cup over 10 probe trials between aCS+, pCS+ (unblocked), pCS- (blocked), and aCS-. Error bars indicate SEM.

The online version of this article includes the following source data and figure supplement(s) for figure 3:

**Source data 1.** Probing vicarious associative learning.
**Figure supplement 1.** Latency to entry for the different groups.

## Blocking the vicarious experience of reward impairs associative learning

Here a similar two-factor repeated measures ANOVA with stimulus type and bin as factors and the time spent in the food trough as the dependent variable was performed for the control group 1 (combined inserted wall and no partner present subgroups). We also found a significant main effect of cue type on time spent in food trough ($F_{(1.372, 20.585)}$=31.215, p<0.001, $\eta_p^2$ = 0.675). Here we find again that the time spent in food trough was higher for aCS+ than aCS- trials (mean difference = 32.043, SE = 2.958, p<0.001; Cohen's d = 1.91, *Figure 3b*). Crucially, pairwise comparisons revealed that the time actors spent in the food trough did not differ for pCS+ cues compared to pCS- cues (mean difference = 3.687, SE = 2.146, p=0.637, Cohen's d = 0.30; *Figure 3b*). We did find a significantly higher responding to the pCS+ cue compared to the aCS- cue (mean difference = 7.948, SE = 1.862, p=0.004, Cohen's d = 0.75), potentially reflecting some non-social appetitive value related to second-order conditioning, while responding to the pCS- cue was not significantly different from the aCS- cue (mean difference = 4.260, SE = 1.634, p=0.119, Cohen's d = 0.44; *Figure 3b*). Additionally, we again found a main effect of bin number on time spent in food trough ($F_{(1.789, 26.834)}$=13.270, p<0.001, $\eta_p^2$ = 0.469) but no interaction between condition type and bin number on the time spent in food trough ($F_{(6.029, 20.585)}$=0.835, p=0.547, $\eta_p^2$ = 0.137).

## Unequal outcomes prevent associative learning

Here a similar two-factor repeated measures ANOVA with stimulus type and bin as factors and the time spent in the food trough as the dependent variable was performed for the unequal outcomes control group. The two-factor repeated measures ANOVA revealed a significant main effect of probe trial type on time spent in food trough ($F_{(1.732, 25.980)}$=46.215, p<0.001, $\eta_p^2$ = 0.755). Again, responding was higher for aCS+ than aCS- trials (mean difference = 30.162, SE = 3.754, p<0.001; Cohen's d = 1.76, *Figure 3c*). Crucially, responding to pCS+ cues did not differ from pCS- cues across the five bins of extinction (mean difference = 0.513, SE = 2.294, p=1.00, Cohen's d = 0.04; *Figure 3c*). No differences were found for pCS+ cue compared to the aCS- cue (mean difference = 0.318, SE = 1.622, p=1.00, Cohen's d = 0.02) or the pCS- cue versus the aCS- cue (mean difference = −0.195, SE = 2.100, p=1.00, Cohen's d = −0.01; *Figure 3c*). As expected, we found a main extinction effect of bin number on time spent in food trough ($F_{(4, 60)}$=8.291, p<0.001, $\eta_p^2$ = 0.356) but found no interaction between condition type and bin number on the time spent in food trough ($F_{(4.528, 67.915)}$=1.671, p=0.160, $\eta_p^2$ = 0.100).

## Socially unblocked cues associated with faster food cup entry than control group cues

Latency scores during the probe trials for the experimental group (combined social-appetitive and social-only subgroups), control group 1 (combined inserted wall and no partner present subgroups) and the control group 2 (unequal outcomes) are shown in *Figure 3—figure supplement 1* (a, b, and c). We ran a bootstrapped analysis of latency differences with N = 5000 iterations per experiment, drawing with replacement from the pCS+ and pCS- trials (according to their N) per iteration and storing the difference in mean latency between these trial type. From these distributions (*Figure 3— figure supplement 1d*) of mean latencies, we assessed whether these distributions differed from zero and whether they differed between experimental groups with Z-tests. We found that the

latency difference scores differed from zero in the experiment group (Z = −2.79, p=0.003) but not control group 1 (Z = −0.90, p=0.18) and control group 2 (Z = 2.08, p=0.98). We furthermore found that the latency difference is bigger in the experiment group compared to the control group 1 (Z = −1.712, p=0.043, one-sided) and control group 2 (Z = −4.72, p<0.001). Finally, control group 1 has larger latency differences than control group 2 (Z = −3.16, p<0.001). We conclude from these results that the rats in the experimental groups showed shorter latencies on pCS+ than on pCS- trials but this was not the case for rats in control group 1 and control group 2. Importantly, this difference is significantly larger in the experimental group than in both control groups, further supporting the interpretation that pCS+ cues acquire associative value in the experimental group, supporting social unblocking.

Taken together, these results show that the actor rats exhibited more food cup directed behavior for the pCS+ cue than both the aCS- and pCS- cue over 10 trials of extinction in the experimental condition only. This means that when actor rats have fully learned a stimulus-reward association producing reward for themselves, adding a cue that predicted an additional reward delivery to a partner rat unblocked associative *social* learning (pCS+>pCS-) about this cue, putatively due to a vicarious reward experience. By contrast, rats did not spend more time in the food cup for the pCS- cue compared to the aCS-, suggesting that additional cues that did *not* predict vicarious reward remained blocked from acquiring associative value. Contrary to the findings for the experimental group, the rats in control groups 1 and 2 did not show such conditioned responding, indicative of acquired value for the unblocked pCS+ cue, over 10 trials of extinction. This suggests that acquiring associative *social* value in this unblocking experiment requires social information exchange (control group: inserted wall) and/or the presence of a partner (control group: no partner present). Interestingly, disadvantageous unequal reward distributions putatively modulated the vicarious reward experience, impeding the unblocking effect. Our results reflect that cues related to mere reward delivery have to be controlled for, as witnessed by the pCS+ over aCS- difference even in the control experiments, highlighting the need for an active blocking control cue (pCS-) as implemented here.

## Strength of the social unblocking effect over trials

We conclude from the simple effects analyses on the interaction effects in the experimental group that the associative value of unblocked novel cues can be measured for approximately six trials in extinction and will use this analysis window going forward. First, we extended the previous analyses with a mixed repeated measures ANOVA design with trial type (aCS+, pCS+ [unblocked], pCS- [blocked], aCS-) and trial 1–6 (bin 1–3) as within-subject factors and group (experimental group 1 vs control group 1 [a and b] vs control group 2) as between-subject factor. Performing this analysis for both 10 s and 30 s period, we found that rats exhibited more food cup directed behavior for the pCS+ cue than both the pCS- cue over six trials of extinction in the experimental group 1 and not control groups 1 and 2 (see *Figure 3—source data 1*).

To directly contrast the unblocking effect between the experimental and control conditions, we calculated difference scores (*Figure 4*) for the direct comparison of the aCS+/aCS-, pCS+/pCS-, pCS +/aCS-, and pCS-/aCS- trial types, and tested for difference in these contrasts between experimental groups. Summary statistics for the *Figure 4* comparisons (F-stats, p-values, effect sizes) can be found in *Figure 4—source data 1*. We would expect no difference in the initial discrimination learning contrast, aCS+/aCS- tested in extinction, between groups. Indeed, in a two-way repeated measures ANOVA, we found no significant main (within-subject) effect of trial number on this contrast ($F_{(4.016, 8.032)}$=5.96, p=0.666) and no interaction effect ($F_{(10,130)}$ = 0.460, p=0.914). Importantly, we did not find evidence for a between-subjects effect of group ($F_{(2, 49)}$=1.859, p=0.167, $\eta_p^2$ = 0.071; *Figure 4a*). The aCS+/aCS- contrast score of the experimental group was not higher than control group 1 (mean difference = 11.242, SE = 6.363, p=0.251) and control group 2 (mean difference = 9.400, SE = 6.363, p=0.438). When directly comparing the unblocked/blocked (pCS+/pCS-) contrast between groups, we *did* expect to find a between-subject group effect. Indeed, when we examined the difference scores of the pCS+/pCS- contrast with a two-way repeated measures ANOVA with group (experimental, control) and trial 1-6 as factors, we found a significant main (between-subject) effect of group ($F_{(2, 49)}$=6.397, p=0.003, $\eta_p^2$ = 0.207; *Figure 4b*). This analysis revealed that the percent difference in responding between pCS+ and pCS- cues was higher for the experimental group than the control group 1 (mean difference = 12.903, SE = 4.531, p=0.019) and control group 2 (mean difference = 14.520, SE = 4.531, p=0.007). We found no within-

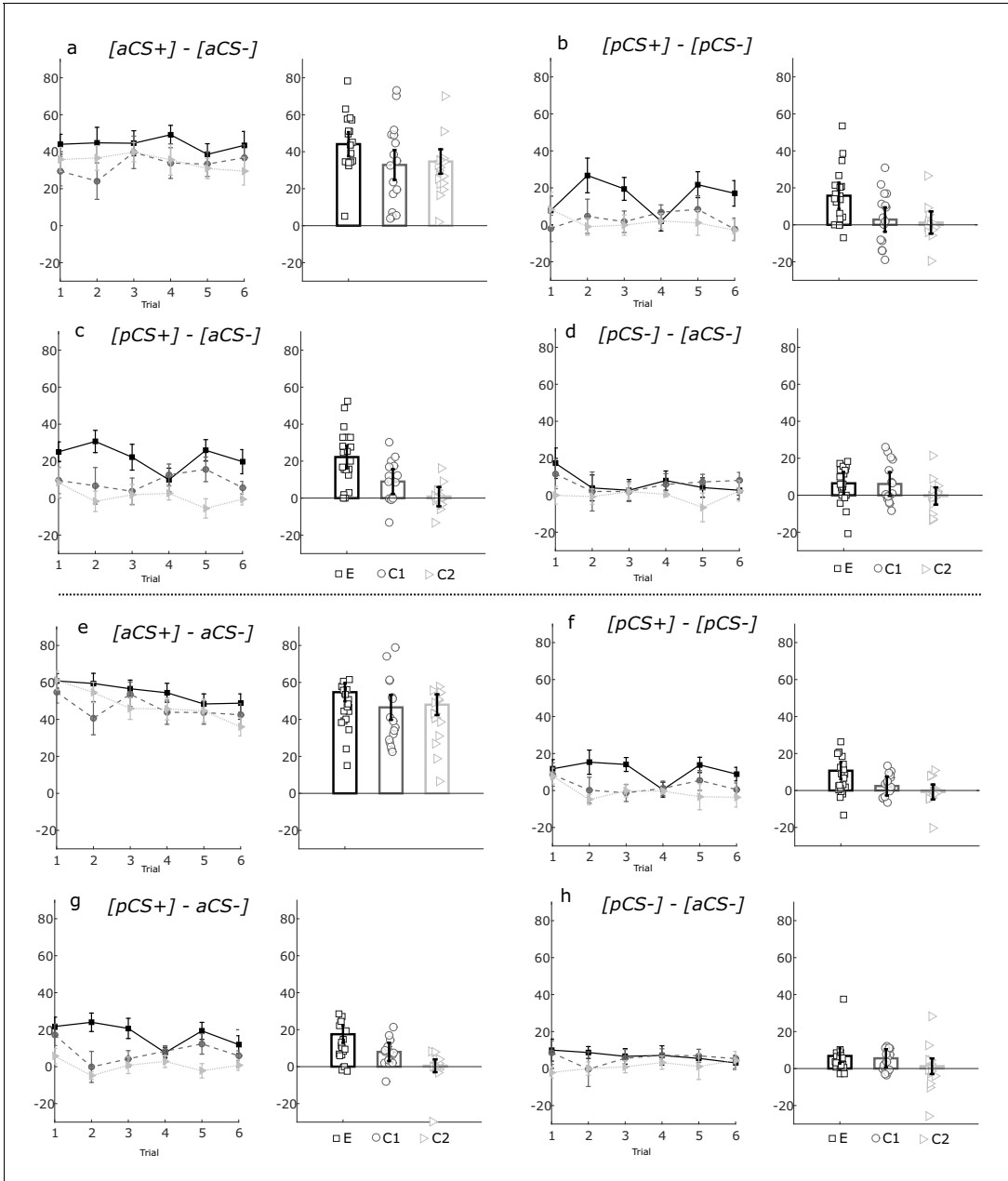

**Figure 4.** Different scores of the actor rats during extinction for 10 s and 30 s data for experimental group (combined social-appetitive and social-only subgroups: E, black squares), control group 1 (combined inserted wall and no partner present subgroups: C1, gray circles) and control group 2 (unequal outcomes: C2, gray triangles). (a, e) Different scores of the percentage of time spent in food cup in the 10 s period (a) and 30 s (e) after cue onset over six trials for the [aCS+]-[aCS-] difference scores. Bar plots show the average over trials of the [aCS+]-[aCS-] difference scores with dots showing the mean per rat. (b, f) Difference of the percentage of time spent in food cup in the 10 s (b) and 30 s (f) period after cue onset over six trials for the [pCS+]-[pCS-] difference scores. Bar plots show the average over six trials of the [pCS+]-[pCS-] difference scores with dots showing the mean per rat. (c, g) Difference of the percentage of time spent in the food cup in the 10 s (c) and 30 (g) second period after cue onset over six trials for the [pCS+]-[aCS-] difference scores. Bar plots show the average over six trials of the [pCs+]-[aCS-] difference scores with dots showing the mean per rat. (d, h) Difference of the percentage of time spent in the food cup in the 10 s (d) and 30 s period after cue onset over six trials for the [pCS-]-[aCS-] difference scores. Bar plots show average over six trials of the [pCs-]-[aCS-] difference scores with dots showing the mean per rat. Error bars indicate SEM.

The online version of this article includes the following source data for figure 4:

**Source data 1.** Strength of the social unblocking effect over trials.

subject effect of trial number ($F_{(4.111, 8.222)}$=0.627, p=0.680, $\eta_p^2$ = 0.013), suggesting that the difference is relatively stable across trials, and no interaction ($F_{(5, 130)}$ = 1.090, p=0.370, $\eta_p^2$ = 0.043). In the contrast analysis for the 30 s period, we also found a significant main (between-subject) effect of group for the pCS+/pCS- contrast ($F_{(2, 49)}$=5.976, p=0.005, $\eta_p^2$ = 0.196; *Figure 4f*), revealing that the percent difference in responding between pCS+ and pCS- cues was significantly higher (one-sided) for the experimental group than control group 1 (mean difference = 8.278, SE = 3.478, p=0.064) and significantly higher than control group 2 (mean difference = 11.493, SE = 3.478, p=0.005). As expected, also for the 30 s period, the aCS+/aCS- revealed no main (between-subject) effect of group ($F_{(2, 49)}$=1.251, p=0.295, $\eta_p^2$ = 0.049; *Figure 4e*). Results from the full contrast analyses for the pCS+/aCS- and pCS-/aCS- can be found in Table S1. We conclude from these results that the pCS+, predicting partner reward in a BR compound, became unblocked and acquired associative value in the experimental group but not in control groups 1 and 2, as witnessed by a significantly larger unblocked versus blocked contrast in the experimental versus control groups for both the first 10 s after cue onset and the whole 30 s period. We attribute this differential social unblocking effect to a putative difference in experienced vicarious reward. The control conditions, impeding social information exchange (control group; inserted wall) and/or the absence of the partner rat (control group; no partner present) presumably attenuated vicarious reward experience. In addition, disadvantageous unequal reward distributions did not lead to unblocking, suggesting that such distributions do not reflect vicarious reward experiences for our rats, in line with previous behavioral evidence of inequity aversion.

## Probing vicarious reward identity

*Burke et al., 2008* found that changing the sensory identity (flavor) of an outcome associated with an added cue in a compound also unblocked this cue and that this identity unblocking was captured by scoring the food cup rate, for example, the frequency or number of entries into the food cup irrespective of the total duration of visits. In our paradigm, social unblocking could also be interpreted as a reward identity switch in that the additional partner outcome changes the sensory aspects of the reward by virtue of the partner receiving and eating the rewards. Food cup rate in the probe trials, next to food cup occupancy, could therefore potentially reflect model-based reward identity unblocking and therefore could provide insight in the influence of sensory features of social unblocking. Alternatively, if the additional partner reward is interpreted solely as a change in value but not identity, we would hypothesize that food cup rate would not be affected.

## Vicarious reward unblocks food cup rate in experimental but not in control groups 1 and 2

To further explore the food cup rate as a measure of the potential identity unblocking effect in the experimental versus control condition, we applied the same a mixed repeated measures ANOVA design with trial type (aCS+, pCS+ [unblocked], pCS- [blocked], aCS-) and bin 1–3 as within-subject factors and group (experimental [N = 20] vs control 1 [N = 16] vs control 2 [N = 16]) as between-subject factor with food cup rate per minute for the first 10 s after the cue onset as dependent variable. We found a significant main effect of trial type ($F_{(1.761, 86.275)}$=78.460, p<0.001, $\eta_p^2$ = 0.616) and an effect of bin ($F_{(1.953, 5.713)}$=21.9968, p<0.001, $\eta_p^2$ = 0.310). We also find an interaction effect of experiment * trial type ($F_{(3.52, 86.275)}$=5.033, p=0.002, $\eta_p^2$ = 0.170). Post-hoc comparison revealed here as well that the food cup rate was significantly higher for pCS+ cue in comparison to the pCS- cue in the experimental group (mean difference = 3.883, SE = 1.207, p=0.014; *Figure 3g*) but not in control group 1 (mean difference = 1.500, SE = 1.350, p=1.00; *Figure 3h*) and not in control group 2 (mean difference = −0.062, SE = 1.350, p=1.00; *Figure 3i*). Interestingly, we furthermore find that the food cup rate for the pCS+ cue is significantly higher than the aCS- in both the experimental group (mean difference = 5.683, SE = 1.074, p<0.001; *Figure 3g*) and control group 1 (mean difference = 5.125, SE = 1.200, p=0.001; *Figure 3h*) but not in control group 2 (mean difference = 0.438, SE = 1.200, p=1.00; *Figure 3i*). These results could potentially indicate that the actor rats had a clear idea that the identity of the US food rewards in BR trials had changed, even though in the inserted wall control group where information exchange is impeded and/or in the no partner present control group where the partner is absent. We can conclude that identity unblocking as measured by the number of entries into the food cup is also present in this task and

that the sensory aspects of the additional partner presence are necessary to associate the novel cue with positive social associative value (BR >OR).

## Probing associative learning in the partner rat

During the compound phase in the experimental group and the control group inserted wall, the partner rat learns to associate another set of outcomes to the compound cues (*Figure 5b,e*): Both reward (BR: pCS+, aCS+), actor reward (OR: pCS-, aCS+) and no reward (NR: pCS-, aCS-) after going through discrimination learning (*Figure 5a,d*). For the partner, learning about the aCS+ cue is thus confounded by being paired with two qualitatively different outcomes: from the perspective of the partner, it represented both a mutual reward outcome and an unequal disadvantageous reward outcome. It is thus likely that the aCS+ cue value would be increased due to the BR associated value but at the same time decreased due to the disadvantageous unequal outcome on OR trials. We tested whether the aCS+, associated with these multiple types of partner-own-reward outcomes, still

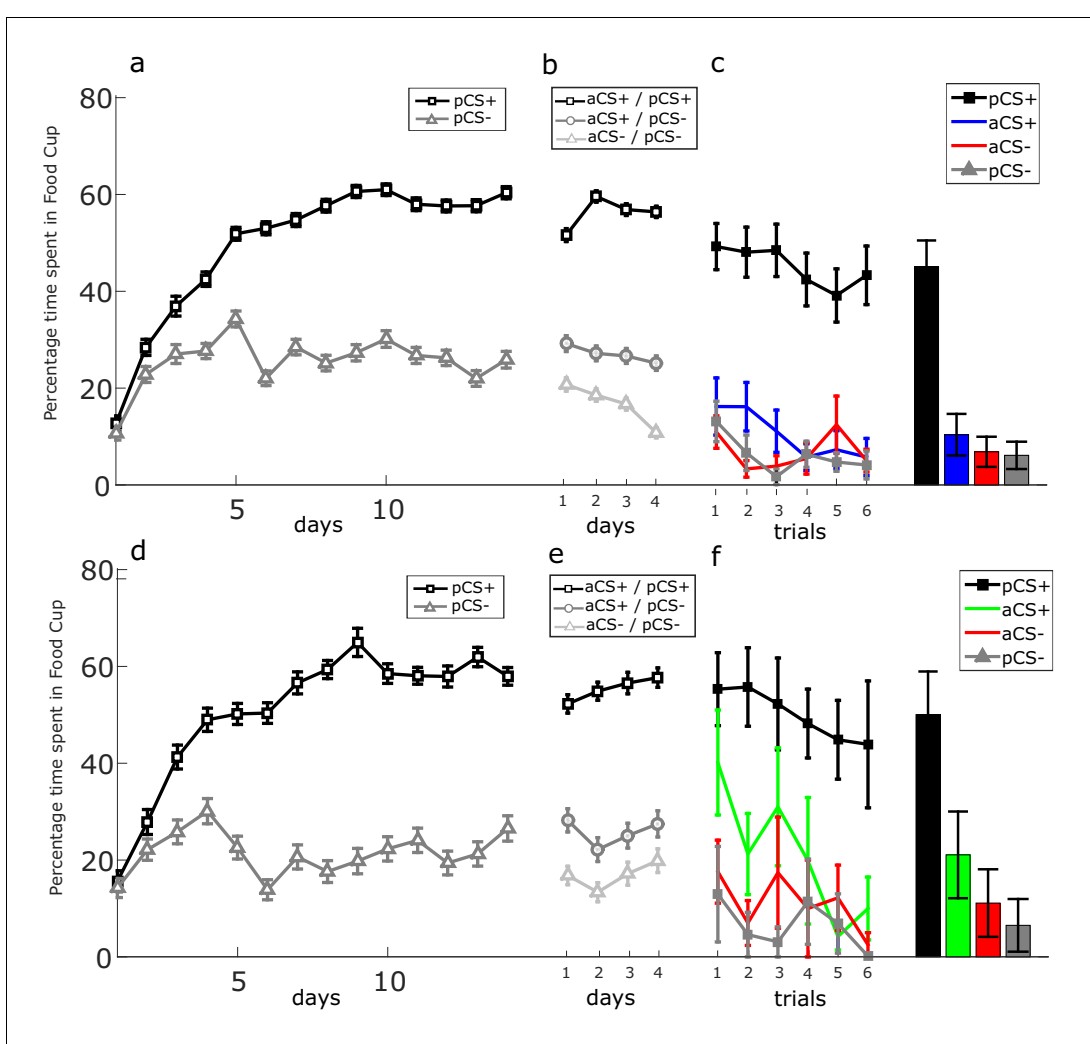

**Figure 5.** Food cup responses of the partner rat during learning. Experimental group 1 (combined social-appetitive and social-only subgroups): (**a**) Percentage of time spent in the food cup during discrimination learning. (**b**) Percentage of time spent in the food cup during social learning. (**c**) Percentage of time spent in the food cup during the first six trials of the probe phase. Control group 1a (inserted wall): (**d**) Percentage of time spent in the food cup during discrimination learning. (**e**) Percentage of time spent in the food cup during social learning. (**f**) Percentage of time spent in the food cup during the first six trials of the probe phase. Bar plots indicate averaged time spent in the food cup over six probe trials between aCS+, pCS+ (unblocked), pCS- (blocked), and aCS-. Error bars indicate SEM.

showed evidence of unblocking by performing a two-factor repeated measures ANOVA with stimulus type and trial 1-6 as factors and the time spent in the food trough as the dependent variable. For the experimental group 1, we found a significant main effect of probe trial condition on time spent in food trough ($F_{(1.551, 29.472)}$=79.840, p<0.001, $\eta_p^2$ = 0.808). As expected, time spent in food trough was higher for pCS+ than pCS- trials (mean difference = 39.00, SE = 3.774, p<0.001, *Figure 5c*), reflecting the partner's initial discrimination learning. However, critically, pairwise comparisons revealed that the partners time spent in food trough was *not* higher for aCS+ cues compared to aCS- cues across six trials (mean difference = 3.530, SE = 1.691, p=0.303; *Figure 5c*). Furthermore, to see whether the absence of social information exchange (inserted wall control group) would influence partner learning we also performed a two-factor repeated measures ANOVA on the data of inserted wall control group. We found a significant main effect of probe trial condition on time spent in food trough ($F_{(3, 21)}$=23.490, p<0.001, $\eta_p^2$ = 0.770). As expected, time spent in food trough was again higher for pCS+ than pCS- trials (mean difference = 43.525, SE = 6.56, p=0.002; *Figure 5f*) here though there was a trend toward higher responding for aCS+ cues than for aCS- cues (mean difference = 9.958, SE = 2.855, p=0.061; *Figure 5f*). We conclude from these results that the compounded cue aCS+ has not become unblocked for the partner rat in the experimental group, however for the control group 1 we observe a clear trend indicative of unblocking. This potential unblocking could be influenced by two factors. First, secondary reinforcement of actor reward delivery (in aCS+ containing compounds) without observation of the actual reward delivery to the actor could have inhibited the attenuating effect of disadvantageous inequity aversion on unblocking and lead to this trend toward unblocking. Next, attentional-based unblocking (*Haselgrove et al., 2013*) could play a role for the partner rat when it learns that one added cue predicts both reward and the omission of reward. This attentional unblocking effect would also be stronger if evidence of actor presence/reward would be blocked. A direct test of unblocking where both rats experienced disadvantageous unequal rewards was implemented as unequal outcomes. We did not include a version of the unequal outcomes control experiment where we also impeded social information transfer but would speculate that, in that case, unblocking would remain supressed as well.

The social unblocking effect persists when controlling for secondary reinforcement but not when the partner is not present. Besides the vicarious experience of reward, other confounding factors could have contributed to learning/unblocking in our paradigm. Most notably, sources of secondary reinforcement should be excluded as potential drivers of learning. During discrimination learning, the actor rat is conditioned to receive pellets contingent on its aCS+. Afterward, in the compound phase, the rat is presented with an auditory–visual compound. Instead of one pellet drop (self-reward), now, on some trials, two pellets drop simultaneously (both reward trials). It is possible that the additional pellet delivery sound acted as a third CS+ in the compound, in addition to the aCS+ and pCS+. Because the sound of the pellet dispenser is already associated with the aCS+ of the actor rat, it is possible that the appetitive value increased with the intensity of this cue (two pellets dropping instead of one), thus enhancing the total value of cue configuration, leading to unblocking of the pCS+. To control for this possible source of secondary reinforcement, in a subgroup of rats (experimental group; social-only), we added a pellet dispenser aimed outside the box (placed at the same location as in the compound phase) already during the discrimination phase, providing the same acoustic features of pellet delivery to the target rat, without presenting additional reward (pellets were collected outside of the box). Next, our control group 1 consisted of a subgroup 1A where a wall impeded social contact and a subgroup 1B where there was no partner present. It is clear that these two groups are similar in that the actor rat does not observe food delivery to the partner. However, in the impeded wall condition, US conditioning could still occur due to pellet dispenser sounds (not controlled for; see *Figure 1—figure supplement 2*) and partner rat related sounds while in the no partner present condition conditioning might still be caused by the observation of food delivery in the other compartment but not pellet dispenser related sounds (controlled for; see *Figure 1—figure supplement 2*). We therefore compare results between these conditions by again looking at responding during the first 10 s of the probe phase, which would equate to time period in which one extra pellet would be delivered. We performed a mixed repeated measures ANOVA design with trial type (pCS+, pCS-) and trial 1-6 as within-subject factors and four group (experimental group; social-appetitive: N = 8; one pellet added vs experimental group; social-only: N = 12; no new pellets added vs inserted wall control: N = 8; one pellet added, vs no partner present control: N = 8; one pellet added) as a between-subjects factor (see also *Figure 1—figure supplement 2*).

We found a significant main effect ($F_{(1, 32)}$=17.964, p<0.001, $\eta_p^2$ = 0.360) of trial type, an interaction of group * trial type ($F_{(3, 32)}$ = 4.559, p=0.009, $\eta_p^2$ = 0.299; *Figure 6a*) and an effect of trial number ($F_{(5, 160)}$ = 7.286, p<0.001, $\eta_p^2$ = 0.186). Post hoc comparison reveals that responding to the pCS+ cue in extinction differs significantly from the pCS- cue in the social-appetitive group (one pellet added; mean difference = 24.808, SE = 4.578, p<0.001), social-only group (no new pellets added; mean difference = 9.717, SE = 4.047, p=0.022) but not the inserted wall group (mean difference = −0.175, SE = 4.956, p=0.972) or the no partner present group (mean difference = 5.525, SE = 4.578, p=0.273). We then also compared responding during the full 30 s, which would equate to the addition of three extra pellets. We performed a similar mixed repeated measures ANOVA design with trial type (pCS+, pCS-) and trial 1-6 as within-subject factors and four group (experimental group; social-appetitive: N = 8; one pellet added vs experimental group; social-only: N = 12; no new pellets added vs inserted wall control: N = 8; one pellet added vs no partner present control N = 8; one pellet added) factors as between-subject factor. We also found a significant main effect ($F_{(1, 32)}$=16.682, p<0.001, $\eta_p^2$ = 0.343), an interaction effect of group * trial type ($F_{(3, 32)}$ = 4.211, p=0.013, $\eta_p^2$ = 0.283; *Figure 6b*) and an effect of trial ($F_{(3.406, 108.98)}$ = 6.084, p<0.001, $\eta_p^2$ = 0.160). Post hoc comparison reveals that the pCS+ cue differs significantly

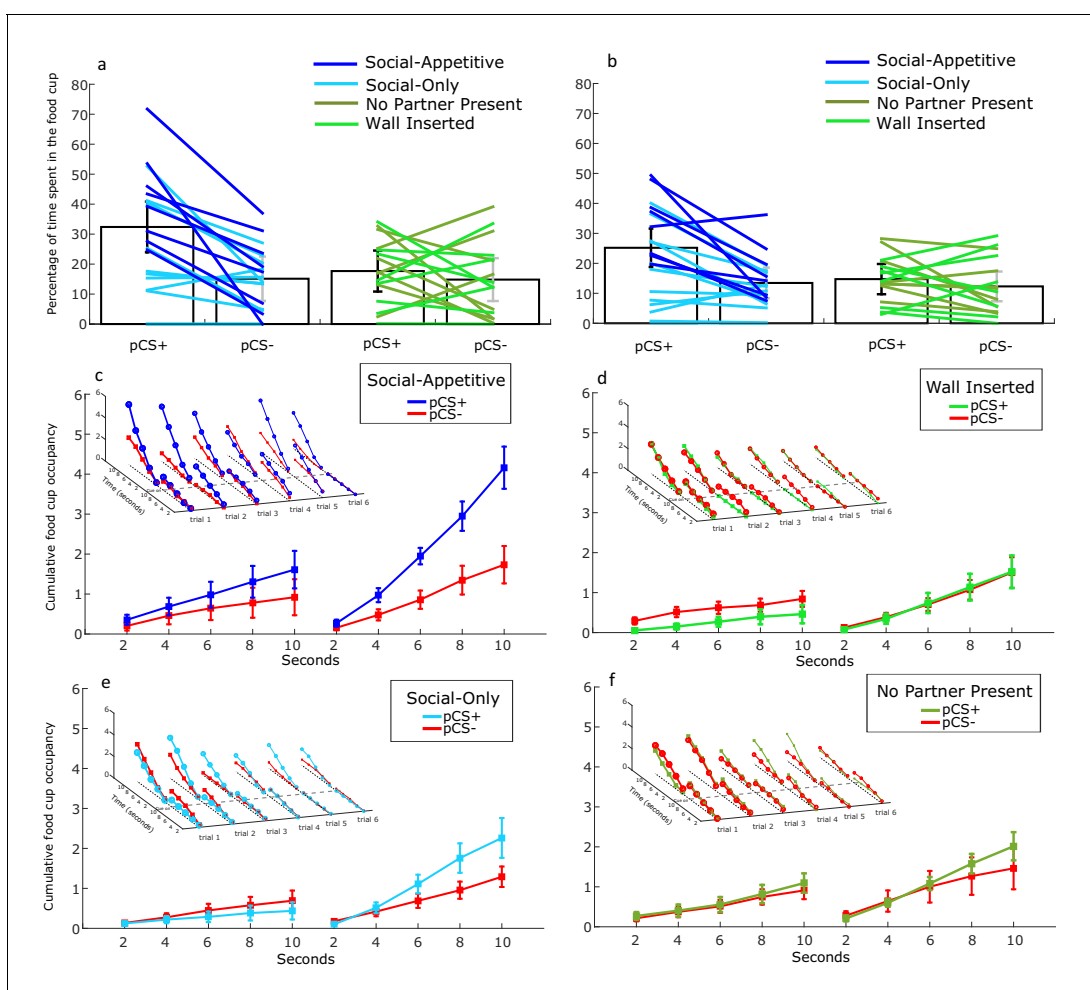

**Figure 6.** Effect of secondary reinforcement, impeding wall and partner absence on food cup occupancy in the first 10 s after cue onset. (a, b) Mean percentage of spent in the food cup, for the 10 s period (a) and 30 s period. (b) After cue onset for pCS+ versus pCS- for trial 1-6 for the experimental group 1 (social-appetitive and social-only) and control group 1 (wall impeded and no partner present). (c) Average cumulative food cup occupancy over six trials for the 10 s pre and post cue on for the social-appetitive group. (d) Average cumulative food cup occupancy over six trials for the 10 s pre and post cue on for the wall inserted group. (e) Average cumulative food cup occupancy over six trials for the 10 s pre and post cue on for the social-only group. (f) Average cumulative food cup occupancy over six trials for the 10 s pre and post cue on for the no partner present group. The 3D plot depicts cumulative food cup occupancy per trial per group. Error bars indicate SEM.

from the pCS- cue in social-appetitive group (one pellet added; mean difference = 16.844, SE = 3.623, p<0.001), in the social-only group (no new pellets added; mean difference = 6.622, SE = 2.58, p=0.032) but not the inserted wall control (mean difference = −1.239, SE = 3.623, p=0.971) and no partner present control (mean difference = 6.106, SE = 3.623, p=0.102). Finally, to zoom in on the temporal dynamics of pCS+ vs pCS- responding in these four experiments, we created time-resolved cumulative occupancy plots. We split the 10 s before and after cue onset in five bins of 2 s each and averaged responding during these bins over the six trials which we found have shown the effect. For an additional statistical analysis, we then looked at the cumulative responding over these five post-cue onset bins. We performed a mixed repeated measures ANOVA design with trial type (pCS+, pCS-) and time bins as within-subject factors and the four groups as between-subject factor. We find a significant main effect ($F_{(1, 32)}$ = 12.601, p=0.001, $\eta_p^2$ = 0.283), an interaction effect of group * trial type ($F_{(3, 32)}$ = 3.780, p=0.020, $\eta_p^2$ = 0.262) and an effect of group * trial type * bin number ($F_{(3.674, 39.674)}$ = 6.084, p=0.025, $\eta_p^2$ = 0.233; *Figure 6c,d,e,f*). Post hoc comparison revealed that cumulative response during the pCS+ cue differs significantly from the pCS- cue in social-appetitive subgroup from bin 2 (mean difference = 0.493, SE = 0.185, p=0.012) and onward on bin 3 (mean difference = 1.091, SE = 0.278, p<0.001), 4 (mean difference = 1.601, SE = 0.385, p<0.001), and 5 (mean difference = 2.427, SE = 0.495, p<0.001) after cue onset. In the social-only subgroup we find significant differences in bin 4 (mean difference = 0.804, SE = 0.314, p=0.015) and 5 (mean difference = 0.972, SE = 0.404, p=0.022). However, no significant temporal bins were found in the inserted wall and no partner present control groups.

While descriptively, the magnitude of the *social* unblocking effect is larger when not controlling for additional pellet drops (social-appetitive subgroup) than when such a control is implemented (social-only subgroup), we conclude that the *social* unblocking effect still exist when explicitly controlling for additional pellets falling in the compound phase for the first 10 s and 30 s period after cue onset but not when social information exchange is impeded and finally also not when no partner is present during the compound phase.

## Discussion

In summary, social valuation is crucial in forming and maintaining social relationships and, presumably, in experiencing the pleasurable and reinforcing aspects of social interaction. However, it remained unclear whether vicarious reward value, which we define here as value derived from social signals associated with reward delivery to another (*Ruff and Fehr, 2014*), could drive learning just as self-experienced value. If this was the case, then vicariously experienced reward should be able to reinforce behavior in a formal Pavlovian learning paradigm. Here we addressed this question by introducing a novel social unblocking task. We find that vicarious reward experience, operationalized in this task as rewards delivered to social partners (cagemates), can indeed drive learning about novel stimuli. After having fully learned that a specific CS+ cue predicted a self-reward, learning about a second cue delivered in compound with this CS+ was blocked, as predicted by learning theory (*Fanselow and Wassum, 2016*; *Rescorla, 1972*), when no additional self or other reward was contingent on this cue. Blocking was found when comparing the food cup response in extinction between trial types, here specifically for the pCS- (CS- cue predicting no partner reward) compared to the aCS- (CS- cue predicting no actor reward). Learning was unblocked, however, by providing an additional reward delivered to the partner simultaneously with the fully predicted self-reward as witnessed by a higher food cup response of actor on the pCS+ (CS+ cue predicting partner reward) compared directly to the pCS-. The social nature of the positive vicarious reward experience was specifically assessed in three control experiments: (1) Preventing the exchange of social information in the compound learning phase impeded the social unblocking effect (pCS+ ≈ pCS-). (2) Partner rat absence during mutual reward showed unblocking for the novel stimuli (pCS+>aCS-) but crucially, not the social unblocking effect (pCS+ ≈ pCS-). (3) When the partner was rewarded and the actor not (aCS-, pCS+) and when the actor was rewarded but not the partner (aCS-, pCS-) we found no evidence either unblocking (pCS+ ≈ aCS-) or social unblocking (pCS+ ≈ pCS-). These results suggest that vicarious reward experience can indeed drive learning processes, in line with formal behavioral learning theory and that specific social aspects of the environment such as partner presence and partner visibility are necessary for observing a social unblocking effect (pCS+>pCS-).

## Learning theory

Our results extend previous work by *Holland, 1984* and Geoffrey Schoenbaum (*Lopatina et al., 2015*; *McDannald et al., 2011*) on unblocking in appetitive Pavlovian conditioning. These authors found that rats, after learning that distinct cues have specific food outcomes, can show unblocking of learning for cues added in compound, when self-rewards were altered by increasing reward value (e.g. an upward shift from one to three pellets) or a change in reward identity (same reward type but a shift in reward features such as flavor). By contrast, learning was blocked when no such reward change occurred (e.g. same reward amount or same identity). According to reinforcement learning theory, the upshift or change in identity led to a discrepancy between the expected reward (one pellet) and the received reward (three pellets), thus producing a reward prediction error. The theory states that if the added cue reliably predicts the increase in reward/identity outcome, it will acquire the value inherent in the reward itself (*Sutton and Barto, 1981*). The main indicator of learning about the value of a (novel) cue in the unblocking paradigm is an increase in the time spent in the food cup in the probe (extinction) phase (*Lopatina et al., 2015*; *McDannald et al., 2011*). Indeed, we observed a higher time spent in the food cup for the cue predicting mutual rewards to the actor rat and its conspecific than to the cue that predicts own-rewards. Taking into account the results of the control experiments, we conclude that the observed enhanced food cup response, that is, the social unblocking effect could be driven by an upshift-related vicarious reward prediction error. The observed social unblocking effect adds to the emerging literature showing that animals attach value to rewards delivered to conspecifics (*Hernandez-Lallement et al., 2016*; *Kashtelyan et al., 2014*) and learn about cues that predict rewards delivered to others. The social reinforcement learning hypothesis (*Hernandez-Lallement et al., 2016*) proposes that integration of social signals expressed by partners can aid in making appropriate decisions in a social context. Evidence for this hypothesis comes from the prosocial choice task (PCT) in which it was found that, rewards delivered to oneself and to a partner are preferred over a reward delivery only to the actor himself in both monkeys (*Horner et al., 2011*) and rats (*Hernandez-Lallement et al., 2015*; *Márquez et al., 2015*). In rats, it was found that this effect was modulated by the behavior displayed by the other rat (*Márquez et al., 2015*) and that this effect was impaired when the partner was replaced by a toy (*Hernandez-Lallement et al., 2015*) or when the display of the partner's preference was impeded. In monkeys it was furthermore found that cues that are associated with reward delivery to another monkey were preferred over cues that were associated with juice delivery to a chair with no monkey in it and this preference was absent, in the non-social condition, when there was only a juice bottle present (*Chang et al., 2011*). We found that social learning occurs when additional reward was delivered to a visible partner but not when preventing the exchange of social information by an opaque wall or when the partner was absent during social learning, providing further evidence that social signals are indeed necessary for learning in social, other regarding, paradigms. A recent study furthermore, shows that macaques increase licking frequency in line with a higher probability of self-reward but decrease their anticipatory licking with increased probability of reward delivery to another monkey (*Noritake et al., 2018*). The authors interpret this decrease of anticipatory licking as an indicator of the negative affect associated with unequal disadvantageous reward pay-outs. Both monkeys and rats have been found to have a distaste for these unequal pay-outs (*Brosnan and De Waal, 2003*; *Oberliessen et al., 2016*). Here we provide similar evidence that in rats tested in our social unblocking paradigm, disadvantageous unequal reward outcomes do not support unblocking of cues that predict reward to the other rat but not oneself. Our finding extend the results of *Rescorla, 1999* who found that changing the outcome of an aCS- cue by adding a cue in compound that predicts self-reward leads to unblocking of that added cue. This contrasts with the lack of unblocking found here, indicating that the observation of reward delivery to the social partner does not have similar reinforcement properties as adding self-reward, possibly due to the negative affect associated with disadvantageous unequal reward outcomes.

Further research is necessary to see whether cues associated with vicarious reward or social reinforcement can also act as a conditioned reinforcer for instrumental responses of rats, as has been found humans (*Lehner et al., 2017*), in a similar way as has been found for appetitive cues (*Burke et al., 2008*; *Kruse et al., 1983*; *Rescorla, 1994*). Finally, it is important to investigate if cues predicting vicarious rewards can guide rats' choices in a social setting. It has been found that rats choose a reward arm in a T-maze that leads to play behavior more than an arm leading to a social

encounter where play was absent (*Humphreys and Einon, 1981*). Furthermore, social play can induce a social place preference (*Calcagnetti and Schechter, 1992*) and rats are willing to lever press for social play reinforcement (*Achterberg et al., 2016*). Our task indicates that the unblocked cue has attained the rewarding properties of social reward and it is therefore likely that a presented unblocked cue would be preferred over a blocked cue when tested in a two-alternative forced choice task. We finally expect that our social unblocking effect depends on the successful transmission of social signals between the actor and partner rat (*Nicol, 1995*; *Hernandez-Lallement et al., 2016*) and that different signal modalities (auditory, visual, olfactory) might contribute and combine in additive or interactive fashion.

## Conclusion

Overall, these data provide evidence that vicarious reward experience can drive associative learning in rats and that the transmission of social cues between rats is necessary for this learning. Further experiments should be conducted to reveal which mode(s) of social information processing are necessary and sufficient to drive unblocking through social value. Overall, our novel behavioral paradigm could be used to further explore how rats learn about value in social contexts and is well suited to probe the neural circuits involved in social reinforcement learning.

# Materials and methods

## Subjects

About 88 male Long Evans rats where housed in pairs of two and kept under an inverted 12:12 hr light-dark cycle, in a temperature (20 ± 2°C), and humidity-controlled (approx. 60%) colony room. All rats had ad libitum access to food, except during the testing period. During behavioral testing, the rats where food restricted (20 g on weekdays and 22 g in the weekend) and maintained on a body weight of about 90% of their free-feeding weight. All testing was performed in accordance with the German Welfare Act and was approved by the local authority LANUV (Landesamt für Natur-, Umwelt und Verbraucherschutz North Rhine-Westphalia, Germany).

## Apparatus

Testing was conducted in four customised PhenoTyper (Noldus Information Technology) behavioral testing boxes (*Figure 1A*) of 45 × 45 × 55 cm$^3$, supplemented with operant devices (Med Associates) and placed inside a custom-made sound- and lightproof ventilated box. The boxes where modified by adding a custom-made Plexiglas separation wall (*Figure 1A*, left panel), which divided the box into two compartments, to allow the training of a pair of rats at the same time. The separation wall was equipped with a sliding door (dimensions: 20 by 20 cm, located at 7 cm from the left side of the Skinner box) and four rectangular interaction windows (*Figure 1A*, left panel; size: 10 × 1.5 cm$^2$) that were positioned exactly in between the door and the wall holding the stimulation devices used for conditioning. Both compartments of the box contained a food trough (Med Associates, ENV-254-CB) positioned in the middle on the right side. The food troughs were adapted in such a way that the detection photobeams were positioned at the entry point of the food trough. The food trough was connected to an automated pellet dispenser (PTPD-0010, Noldus Information Technology) that delivered sucrose pellets (20 mg dustless precision pellets, Bio-Serv, Germany). Operant devices were positioned on the right side of the box at the level of the separation wall: an LED Stimulus Light (Med Associates, ENV-211m) with green cover was positioned 10 cm above the ground and a house light (Med Associates, ENV-215m) 28 cm above the ground. A speaker (Med Associates, ENV-224am) was positioned at 20 cm above the ground for the playback of auditory stimuli (*Figure 1A*, right panel). Auditory stimuli were played back at a loudness of 75 dB measured with a hand-held analyser (type 2250s from Brüel and Kjaer) right in front of the speaker. In the top cover of the Skinner boxes, a camera (Basler, acA1300-60gc, GigE) was positioned to obtain videos of the behavioral experiment at 25 fps. Analyses of the recorded videos was performed with EthoVision XT 11.5 (Noldus Information Technology). Finally, a USV-microphone was positioned next to the camera for recording ultrasonic vocalizations using Ultra Vox XT (Noldus Information Technology).

## Behavioral training

### Pavlovian discrimination task

Before the start of behavioral training, rats were put on food restriction to reduce their weight to 90% of their free-feeding weight. Within a pair of cage mates, one rat was assigned at random as the actor animal, and the other as the partner animals. As a first step, they were habituated to their pre-determined training side of the customised PhenoTyper for 3 days (15 min per day). During this period, they could retrieve six pellets that were put along the edges of their respective side of the box. Subsequently, the discrimination learning phase started. Here the pairs of rat cage mates were divided into two groups; one group of rat pairs would learn a visual discrimination problem, and the other an auditory discrimination problem (*Figure 1B*, left panel). The visual stimuli to be discriminated consisted of a houselights flashing at 1 Hz (0.1 s on, 0.9 s off) and a steady green light; the auditory stimuli were made up by a 4.0 kHz clicker (0.1 s on, 0.9 s off) and a 1.5 kHz (75 dB) steady tone (see *Figure 1—figure supplement 2* for overview of stimulus contingencies). The different groups (auditory vs visual) were each trained alone either in the upper or lower compartment of the Skinner box, and the side assignments between actor and partner rats were counterbalanced between experiments (*Figure 1A*). Each rat received 14 days of discrimination training. One daily session consisted of 40 trials, of which 20 trials were aCS+ and 20 aCS-. The order of aCS+ and aCS- trials was pseudo-randomized, with no more than three trials of one kind occurring in a row. Stimuli were presented for 30 s and at every 10 s (+ 0.1 to 0.4 s jitter), a pellet was delivered (*Figure 1C*). We trained a total of N = 20 actor rat and 20 partner rats on the discrimination problem in the experimental group, 16 actors and 16 partners in the control group 1 and 16 animals (all considered actors) in control group 2 (unequal outcomes). The experimental group was divided in subgroup 1A (social-appetitive subgroup) and 1B (social-only subgroup) and in subgroup 1A (inserted wall) and 1B (no partner present; See *Figure 1—figure supplement 1* for overview). The social-only subgroup consisted of 12 actors and 12 partners of the N = 40 experimental group. Here a second pellet dispenser was already placed outside of the behavioral box during the entire discrimination learning phase, at the opposite side of where the current rat was trained, delivering pellets outside of the box. This additional dispenser placement ensured that the sound of additional pellet drops was similar to the compound conditioning phase (see below). Providing a uniform pellet delivery sound associated with self-reward pellet delivery throughout the experiment prevented any difference in pellet delivery related sounds as a source of secondary reinforcement from influencing the conditioning to added cues in the compound phase. In the additional social-appetitive subgroup, the second pellet dispenser was not active during the discrimination phase (*Figure 1—figure supplement 1*). This gave us the opportunity to make direct comparison within the experimental group to investigate potential effects of secondary reinforcement (*Figure 1—figure supplement 1*). The ITI in both experimental and control group was made up of a fixed 30 s window supplemented with a randomized time window ranging from 5 to 100 s with steps of 5 ms, uniformly distributed. The ITIs were thus fully randomized, resulting in a total variable ITI with a mean of 80 ms. Ultrasonic vocalizations where recorded from 10 s before cue onset to 20 s after cue offset, for a total duration of 60 s per trial. After completion of the discrimination phase, rats progressed to the compound conditioning stage.

### Compound conditioning

After discrimination training was completed, rats in the visual discrimination group received 1 day of pre-exposure to the two novel auditory stimuli while the rats in the auditory discrimination group received 1 day of pre-exposure to the two novel visual stimuli. The pre-exposure session consisted of one session with six trials. The stimuli were presented in a randomized order with ITIs of 15, 30 45, 60, 75, and 90 s. Pre-exposure was done to minimize an influence of novelty induced enhancement on the conditioning of added compound stimuli (*Holland and Gallagher, 1993*) and enhance the discriminability of these added stimuli (*Honey and Hall, 1989*) for each group. This would strengthen the evidence that any observed blocking or unblocking would be related to task conditions, rather than novelty. In the compound phase, three different conditions were used (*Figure 1B*, middle panel). In both reward (BR) trials, both the (respective visual or auditory) CS+ of the actor group (aCS+) and the partner group (pCS+) were simultaneously displayed and both rats were rewarded with three pellets. In the own reward (OR) trials, the respective aCS+ was simultaneously

displayed with the aCS- of the partner group and only the actor group was rewarded. In the NR trials, the respective aCS- was simultaneously displayed with the aCS- of the partner group and neither actor nor partner were rewarded. A compound conditioning session consisted of 20 trials per condition. The conditions BR, OR, and NR were pseudo-randomized with every condition not being repeated more than three times in a row. ITI randomization, stimulus presentation, and reward delivery were implemented as in the discrimination phase.

### Probe trials

During probe trials, all rats were tested in isolation for one extinction session in their assigned box compartment. All stimuli were now presented in isolation, both the aCS+ and aCS- learned in the Pavlovian discrimination task as well as the two novel stimuli pCS+ (both reward CS+) and pCS- (own reward CS-) added in the compound phase, for which learning was hypothesized to become unblocked and blocked, respectively. Rats in both groups went through 10 trials for each of these four stimuli, presented in isolation and without reward delivery (*Figure 1B*, right panel). The four stimuli were pseudo-randomized with every condition not being repeated more than three times in a row.

### Control experiments

In the inserted wall control experiment, eight actor rats and eight partner rats went through the same three experimental conditions. The only difference here is that during the compound phase, the wall that separated the Skinner box compartments was rendered opaque by adding an additional black wall, to block contact between the actor and partner rats. We hypothesized that if visual, and/or auditory and/or olfactory contact between the rats facilitated the social information transmission that helps to unblock reinforcement learning of compound cues, then obstructing these transmission cues should impair unblocking. In the inserted wall control group, we chose to also implement the 1-pellet dispenser condition (see *Figure 1—figure supplement 1*), to match our results to the condition where secondary reinforcement might still play a role. If differences between the inserted wall control and the social-appetitive experimental conditions would still emerge, this would strengthen the interpretation that social unblocking was driven primarily by vicarious reward, and not by secondary reinforcement learning, as the putative reinforcing effect of an additional pellet drop during the compound phase was present in both the social-appetitive experimental and inserted wall control conditions.

In the no partner present control experiment, eight actors went through the same experimental conditions. Here, the only difference was that during the compound phase the partner rat was not present. Instead, pellet dispensers dropped pellets in a custom-made 3D printed plastic food cup including the metal parts which were used in the original food cup for catching the pellet. This made sure that the sound of pellet delivery was similar as in the experimental group. Pellets furthermore fell through the custom-made food cup in a small cup underneath as to avoid the pellets to stack up in view of the actor rat. Finally, it is important to note that secondary reinforcement of the additional pellet dispenser activity itself was controlled for by delivering pellets outside of the box during discrimination learning, as in the social-only experimental subgroup (see *Figure 1—figure supplement 1*). This would ensure that only the pellet delivery related sound of falling in the food cup (of the empty partner side) and not sounds made by the pellet dispenser itself would influence associative learning. This control condition was used to assess if visual and auditory observation of pellet delivery in the food cup could unblock learning by itself. Finally, in control experiment 2: unequal outcomes, actor and partner rats went through the same stages of conditioning only now during compound conditioning the BR condition became a partner reward condition while the OR remained the same. With this symmetric implementation, actor rats' OR is partner rats' PR and vice versa and both groups of rats can be treated as actors, doubling the sampling size for one experiment. This control experiment was used to assess if disadvantageous unequal reward outcomes to partner rats would unblock learning.

### Statistical data analyses

Entries into the food trough were recorded as photobeam breaks. Raw data were processed in Etho-Vision XT 11.5 (Noldus Information Technology) to extract our dependent variables: time spent in

the food trough and number of entries in the trough (food cup rate). Food cup directed behavior in the form of time spent in the food trough and latency to entry were analyzed per trial and per condition for all stages of learning; further analysis and graph preparation was performed using custom-made scripts in MATLAB (version 2014b, MathWorks). All statistics was performed using SPSS (IBM Corp. Released 2017. IBM SPSS Statistics for Windows, Version 25.0. Armonk, NY: IBM Corp). To assess the strength of learning during discrimination and compound conditioning, only the first 10 s of the cue period was analyzed to avoid the influence of reward delivery/omission feedback (*McDannald et al., 2011*) and the time spent in the food cup was used as measure for conditioned responding. In the probe trials however reward was absent, therefore here we analyzed both 10 s and 30 s period. Previously *Burke et al., 2008* and *McDannald et al., 2011* used the percentage of time spent at the food trough and the food cup rate to assess value unblocking and identity unblocking, respectively as measures for conditioned responding. As social unblocking is thought to mainly reflect value unblocking, we report the percentage of time spent as our main outcome parameter. For completion, we also report food cup rate to assess identity unblocking. Discrimination learning performance was quantified by averaging responding to the cues over the last 4 days of training and comparing the mean between aCS+ and aCS- and difference scores of CS+ - CS- for contrasting cue modalities using paired sample t-tests. Performance in the compound phase was quantified using a two factor repeated measures ANOVA on the mean response rate per day across conditions (BR, OR, and NR) and post hoc tests were performed to assess the significance of any differences between conditions, corrected for multiple comparisons. Performance in the probe trials was assessed by averaging responding of the actor rats time spent in the food cup and food cup rate over five bins (two trials per bin) and running a two factor repeated measures ANOVA over these bins and the four stimuli types (aCS+, pCS+ (unblocked), pCS- (blocked) and aCS-) separately for experimental and control experiments. Differences between conditions and bins were assessed with post-hoc tests, again corrected for multiple comparisons. A putative difference between latencies to entry was analyzed in a two-step process, as latencies were not normally distributed. Using a bootstrap procedure (N = 5000 iterations), per experimental condition we sampled N probe-trial latencies to entry (with N resampled with replacement, equal to the number of trials with valid entries excluding non-entries and latencies < 0.040 s) for the unblocked cue and the blocked cue throughout all probe trials, and stored (per iteration) the difference in mean latency for these samples. This generated an N = 5000 bootstrap population of mean latency differences per experiment, with all of these distributions following a normal-like distribution (see *Figure 3—figure supplement 1*). Using a Z-test, we assessed (1) whether each distribution was significantly different from 0 (suggesting a significant difference in latency to enter between trial types) and (2) whether this latency difference was significantly different between experimental conditions. For the direct comparison between Experiment and control group in the probe trials we performed a mixed repeated measures ANOVA with factors trials (trial 1 to 6) and stimuli (aCS+, pCS+ [unblocked], pCS- [blocked] and aCS-) and experiment group as a between-subjects factor (experiment, control). Differences between conditions between experiments were assessed with post-hoc tests, corrected for multiple comparisons. Finally, to further in depth look at the difference between experiment and control group difference scores were calculated for every available contrast (aCS+/aCS-, pCS+/pCS-, pCS+/aCS-, and pCS-/aCS) and for these contrasts a two factor repeated measures ANOVA was calculated. For all RM-ANOVA's, Mauchly's test of sphericity was performed and, when significant, the Greenhouse-Geisser correction was applied.

## Additional information

### Funding

| Funder | Grant reference number | Author |
|---|---|---|
| Volkswagen Foundation | AZ88216 | Sander van Gurp<br>Jochen Hoog<br>Marijn van Wingerden |

The funders had no role in study design, data collection and interpretation, or the decision to submit the work for publication.

## Author contributions
Sander van Gurp, Conceptualization, Data curation, Software, Formal analysis, Validation, Investigation, Visualization, Methodology, Writing - original draft, Writing - review and editing; Jochen Hoog, Investigation; Tobias Kalenscher, Resources, Writing - review and editing; Marijn van Wingerden, Conceptualization, Formal analysis, Supervision, Funding acquisition, Methodology, Project administration, Writing - review and editing

## Author ORCIDs
Sander van Gurp ![ORCID] https://orcid.org/0000-0003-2265-8156
Marijn van Wingerden ![ORCID] https://orcid.org/0000-0002-0558-6624

## Ethics
Animal experimentation: All testing was performed in accordance with the German Welfare Act and was approved by the local authority LANUV (Landesamt für Natur-, Umwelt und Verbraucherschutz North Rhine-Westphalia, Germany).

## Decision letter and Author response
Decision letter https://doi.org/10.7554/eLife.60755.sa1
Author response https://doi.org/10.7554/eLife.60755.sa2

# Additional files
## Supplementary files
• Transparent reporting form

## Data availability
Dryad data deposited at https://doi.org/10.5061/dryad.v9s4mw6qp.

The following dataset was generated:

| Author(s) | Year | Dataset title | Dataset URL | Database and Identifier |
|---|---|---|---|---|
| van Gurp S, Hoog J, Kalenscher T, van Wingerden M | 2020 | Vicarious reward unblocks associative learning about novel cues in male rats | https://doi.org/10.5061/dryad.v9s4mw6qp | Dryad Digital Repository, 10.5061/dryad.v9s4mw6qp |

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
