## [Decision Letter]

**Decision letter after peer review:**

[Editors’ note: the authors submitted for reconsideration following the decision after peer review. What follows is the decision letter after the first round of review.]

Thank you for submitting your work entitled "Vicarious reward unblocks associative learning about novel cues in male rats" for consideration by *eLife*. Your article has been reviewed by 3 peer reviewers, and the evaluation has been overseen by a Reviewing Editor and a Senior Editor. The following individuals involved in review of your submission have agreed to reveal their identity: Patricia H Janak (Reviewer #3).

Our decision has been reached after consultation between the reviewers. Based on these discussions and the individual reviews below, we regret to inform you that your work will not be considered further for publication in *eLife*.

The study was judged to be a creative and unique attempt to demonstrate a fundamental learning property in a social situation. However the importance is based on demonstrating that it was the social or vicarious nature of the reward and it was judged to be missing significant controls to rule out a variety of other possibilities that are too numerous to complete in within the timeframe of a *eLife* revision.

Reviewer #1:

This study uses classic learning theory procedures – blocking and unblocking – to determine if learning the relationship between a pavlovian stimulus and reward can occur through observation. The authors suggest that cues that predict reward to a conspecific can 'unblock' learning and cause observer rats to respond to a stimulus as if they expect reward for themselves. It is an interesting and clever idea but I have several major problems with the study.

1) The authors do not show that they actually get blocking. Overall, I thought the results would have been more significant if the authors showed that vicarious reward could block new learning, but instead the authors focus on unblocking. The problem is that they don't actually demonstrate that they would get blocking in their paradigm. Thus, I'm not convinced that vicarious reward can unblock learning when there is no evidence of blocking to begin with. Also, with unblocking it seems like anything that changes attention and arousal might cause a stimulus to become associated with physical reward that the animals gets during those compound stimuli.

2) I think the authors are missing a critical control where reward is delivered to an empty box or fake rat as done in most primate studies. It could be that just having a pellet delivered to the food cup on the other side of the box would produce unblocking.

3) There is no analysis of video. It is not clear what the rats are paying attention to during the task. Do the rats pay attention to each other? During the control sessions do rats investigate the barrier to gain access to the other rat or are they completely unaware that there is a rat on the other side of the box? Do these measures vary across contexts and stimuli?

4) We are told very little about the behavior of the other rat. How well did they respond to cues during social and non-social tasks? Did obtaining food bring the partner rat closer to the central part of the cage so that the rats interacted more? If so, this could have served as a social reward. That is, during 'own' reward trials the partner rat would not be visible to the target rat thus this stimulus would be less rewarding than 'both' reward trials. During 'both' reward trials, the target rat received two rewards, one is the pellet and the other is to see and possibly interact with the other rat.

5) The fact that the pellet dispenser worked so well as a secondary reinforced made the results less compelling. Was the pellet dispenser outside the box (control) as audible to the observer rat as when it was inside the box? Also, is it possible the clanging of the pellet acted as a cue? These issues are another reason why the empty box control mentioned above seems critical.

Reviewer #2:

The submission by van Gurp et al., introduced a behavioral model to show that exposing rats to a social partner during a pavlovian task can drive and maintain reinforcement learning. The authors used a classical blocking/unblocking procedure to train rats first to discriminate between two stimuli (either visual or auditory) that predicted either a food pellet delivery (CS+) or nothing (CS-). In a second phase of the experiment, the authors exposed the rats to a compound training during which both rats (actor and partner) were trained together in the same box divided in two compartments by a transparent barrier. During this phase, the actor rats were exposed to either (1) aCS+ and pCS+ (actor and partner related cues – under this schedule both rats received a pellet); (2) aCS+ and pCS- (only the actor rats received a pellet); or (3) aCS- and pCS- (no pellets for both rats). Finally, the authors presented each cue separately in an extinction probe test showing that once actor rats have fully learned a stimulus-reward association producing reward for themselves, adding a cue that predicted an additional reward delivery to a partner rat unblocked associative learning about that cue. Moreover, by adding an opaque barrier between the two sides of the chamber and preventing visual contact, the authors showed that the pCS+ did not unblock associative learning.

I enclose below my comments for the authors.

1) The behavioral work, while intriguing, is under-developed. It is not completely clear what is the role of the social partner in this task. The authors should report the behavioral data for this group for two major reasons. Firstly, in a pavlovian task like the one described in this manuscript, there is not a real distinction between actor and partner. Indeed, each actor is the partner of the other rat and vice-versa. Secondly, and probably more importantly, reporting the data of the social partners can be a useful way to show replication of the data (which is a critical feature of a new behavioral procedure). Moreover, the authors should report more data describe the rats' behavior during the task (i.e. is the actor rat watching the partner consuming the pellets? Do both rats enter the food cup at the same time? etc.).

2) In line with the above point, it seems that the unblocking effect is driven by the simple presence of a social partner. Indeed, the simple presentation of pCS+ without the visual contact of the social partner, did not unblock associative learning. However, it is hard to disentangle the role of pCS+ from the presence of the social partner only. Therefore, the authors should add a control group in which only the social partner is getting a pellet (aCS- and pCS+). The authors reported some rational to explain why they did not run this experiment; however, this group can show that even without receiving a pellet the pCS+ has a strong effect on the actor rats during the unblocking procedure. This will show a real social transmission effect from the social partner to the actor.

3) It is also critical to replicate the finding from the Control group. Given the lower n (8 vs 20 rats in the Experimental group), the variability in this group is bigger than the first experiment and also the magnitude of the effect (for both aCS+ and aCS-) is different relative to the experimental group. This is a major concern considering the reliability of the new model.

4) The direct comparisons between experimental and control conditions seems slightly strange. It is hard to direct compare two groups that have been exposed to different behaviors. One group experienced the presence of a social partner and the other did not. I suggest avoiding this statistical comparison, also because the author decided to arbitrarily select 6 trials for this comparison instead of the 10 total trials. Moreover, for the control group the authors reported no interaction between condition type and bin number (Subsection “Control group”), but still reported posthoc comparison. Please clarify.

5) The authors should present the data of the auditory vs visual cues in two different lines (and different colors) in the same graph before collapsing them in a single curve. It is important to see that the two modalities are similar.

Reviewer #3:

van Gurp and colleagues have conducted an innovative experiment on possible learning by observer rats of cue-reward associations when that reward is delivered to a nearby familiar cagemate. Using the blocking procedure, during compound cue presentation reward is given as before to the observer rat, and, at the same time, delivered to a cagemate directly beside the observer rat in a clever chamber that should allow for visual, olfactory, and auditory information sharing. The results show that the observer rat responds more for the newly added cue that was paired with cagemate reward than for a newly added cue paired with no cagemate reward, but paired with the expected reward (traditional blocking condition). A control group failed to show clear differences in responding between the observance unblocked cue and the traditionally blocked cue when observational across subjects was presumably greatly reduced by insertion of a divider between testing cage compartments.

These findings were well-predicted by the foregoing literature, and are an elegant test of the power of vicarious reward to impact learning. The findings provide evidence that vicarious reward acts similarly as directly experienced reward, adding to the body of work extending reward learning mechanisms to aspects of social interaction.

I found no major shortcomings of this work. It is the case that one measure, percent time in the food cup, provided the strongest evidence for the unblocking, while the others are less clearly supportive. I do not view this as a shortcoming since multiple measures of conditioned behavior could tap into distinct aspects of the associative process. I do wonder if it might be more convenient if the authors move the supplemental data to the main manuscript so it is easier to inspect the findings.

One might expect more content here, such as a neurobiological follow-up study. While this work sets the stage for that, the initial behavioral demonstration in my view warrants publication.

[Editors’ note: further revisions were suggested prior to acceptance, as described below.]

Thank you for submitting your article "Vicarious reward unblocks associative learning about novel cues in male rats" for consideration by *eLife*. Your article has been reviewed by Kate Wassum as the Senior Editor, a Reviewing Editor, and three reviewers. The following individual involved in review of your submission has agreed to reveal their identity: Matthew Roesch (Reviewer #2).

The reviewers have discussed the reviews with one another and the Reviewing Editor has drafted this decision to help you prepare a revised submission.

Reviewer #1:

The authors addressed my requests and, by adding the requested control groups, they improved the manuscript relative to the first submission.

Reviewer #2:

The authors addressed all my concerns by adding in the critical controls. They state they will do USV and video analysis later. I'm fine with this because of the additional controls address my concerns. I think it is a very interesting study.

Reviewer #3:

This is a revised manuscript that I previously reviewed. I liked the original paper, and I maintain that opinion now. I think the possibility that unblocking in an observing rat occurs following pairing the 'new' cue during compound training with reward received by a neighbor that can be seen, heard, and smelled is of interest. This work is likely to stimulate further study. The authors have strengthened the paper, and the original findings are maintained.

The degree of responding to the 'new' cue in the control group that can't see or interact through holes with their cagemate is greater than a condition in which there is no second rat, only free food delivered to an empty chamber. This suggests some amount of learning about the reward receipt of the partner rat may still be occurring (vocalizations?) but this doesn't dampen my interest in this work.

---

## [Author Response]

[Editors’ note: the authors resubmitted a revised version of the paper for consideration. What follows is the authors’ response to the first round of review.]

Reviewer #1:This study uses classic learning theory procedures – blocking and unblocking – to determine if learning the relationship between a pavlovian stimulus and reward can occur through observation. The authors suggest that cues that predict reward to a conspecific can 'unblock' learning and cause observer rats to respond to a stimulus as if they expect reward for themselves. It is an interesting and clever idea but I have several major problems with the study.1) The authors do not show that they actually get blocking. Overall, I thought the results would have been more significant if the authors showed that vicarious reward could block new learning, but instead the authors focus on unblocking. The problem is that they don't actually demonstrate that they would get blocking in their paradigm. Thus, I'm not convinced that vicarious reward can unblock learning when there is no evidence of blocking to begin with. Also, with unblocking it seems like anything that changes attention and arousal might cause a stimulus to become associated with physical reward that the animals gets during those compound stimuli.2) I think the authors are missing a critical control where reward is delivered to an empty box or fake rat as done in most primate studies. It could be that just having a pellet delivered to the food cup on the other side of the box would produce unblocking.3) There is no analysis of video. It is not clear what the rats are paying attention to during the task. Do the rats pay attention to each other? During the control sessions do rats investigate the barrier to gain access to the other rat or are they completely unaware that there is a rat on the other side of the box? Do these measures vary across contexts and stimuli?4) We are told very little about the behavior of the other rat. How well did they respond to cues during social and non-social tasks? Did obtaining food bring the partner rat closer to the central part of the cage so that the rats interacted more? If so, this could have served as a social reward. That is, during 'own' reward trials the partner rat would not be visible to the target rat thus this stimulus would be less rewarding than 'both' reward trials. During 'both' reward trials, the target rat received two rewards, one is the pellet and the other is to see and possibly interact with the other rat.5) The fact that the pellet dispenser worked so well as a secondary reinforced made the results less compelling. Was the pellet dispenser outside the box (control) as audible to the observer rat as when it was inside the box? Also, is it possible the clanging of the pellet acted as a cue? These issues are another reason why the empty box control mentioned above seems critical.

We first address the comments of reviewer 1. This reviewer comments that we do not show the blocking effect in our paradigm and therefore stimuli cannot become unblocked in our task. We agree and value the importance of showing blocking and now include argumentation supporting that we do indeed observe blocking in our task. In both the experimental group, control group 1 and control group 2, the aCS+ was fully learned and coupled to novel cue predicting no reward to the partner rat (pCS-, in the “Own Reward” [OR] compound). Consequently, blocking should be observed in relation to the pCS- cue, because there are no changes in self- or partner-reward from the perspective of the actor rat. To show blocking in our task setup, we directly compare the novel pCS- cue to the fully learnt aCS- cue. We found no significant differences between the pCS- and aCS- across all groups, supporting our interpretation that learning about this cue was blocked (see Figure 3A,B and C).

We do acknowledge that, in early trials, there is an enhanced response to the pCS- relative to the aCS-. We think however that this could be caused by some learning driven by social facilitation during the compound phase: there, partner rats react more to their own pCS- learnt cue when it is presented in the pCS-/aCS+ compound (OR for actor, a PR trial from the perspective of the partner), than when it is presented in the aCS-/ pCS- compound (NR; See new figure S1B). If actor rats interpret the partner's approach to the food cup during OR trials as a signal that the partner rat is rewarded, this could putatively lead to some unblocking. However, in the new control experiment 2, where both rats are only rewarded alternatingly (OR and PR trials), complete blocking can be found (Figure 3C,F,I). This enhanced probe-phase response to the pCS- in the experimental condition the decreases rapidly though, while the pCS+ related unblocking response last until trial 6 (cf. Figure 4B vs 4D).

Reviewer 1 furthermore comments on the fact that we miss the empty box control as present in most primate studies. We agree that this control is critical and we have therefore implemented their comment as control experiment 1b. There, we show that the social unblocking effect of differential responding for pCS+ > pCS- is absent and therefore conclude that the vicarious reward experience of mutual reward delivery to actor and partner is necessary to drive social learning in our task. We do observe more responding to the pCS+ over the aCS-, which we attribute to the influence of secondary reinforcement of food related cues that we have shown to influence the strength of the social unblocking effect (Experimental group 1b).

Reviewer 1 also commented on the influence of secondary reinforcement in our experiment. We indeed show that unblocking is smaller when explicitly controlling for pellet delivery sounds. This indicates that the sounds of pellet delivery can indeed augment the social unblocking effect, but our experiments show that this secondary reinforcement interpretation cannot solely explain our results. We would like to address this comment by going through the observed findings of the different control experiments. Firstly, when not controlling for secondary reinforcement but impeding social information exchange, we do not observe unblocking, indicating that the audible additional pellet delivery sounds alone cannot account for the observed unblocking. Furthermore, when we do control for pellet dispenser sounds in control experiment 1b (by running a second pellet dispenser already during discrimination learning) but do drop pellets in an empty chamber with no partner rat present, it also did not lead to social unblocking (pCS+ > pCS-). Finally, we would like to add that observing a partner rat being fully rewarded in control group 2 (pCS+/aCS-, PR), including pellet delivery sounds and pellet consumption, still does not lead to unblocking. Summarised, these control experiments show that partner rat presence and mutual reward outcomes are necessary to drive our reported social unblocking effect.

Finally, Reviewer 1 also comments on the importance of social interaction between rats during cue presentation. We agree completely with this and are running active analyses on not only social interaction scoring, but also ultrasonic vocalisation detection and classification in the context of this social task. We hope that these extensive analyses should help elucidate how social learning and social transfer of reinforcement value takes place and aim to present them in a future manuscript as it is beyond the scope of the current paper that establishes the behavioral paradigm.

Reviewer #2:The submission by van Gurp et al., introduced a behavioral model to show that exposing rats to a social partner during a pavlovian task can drive and maintain reinforcement learning. The authors used a classical blocking/unblocking procedure to train rats first to discriminate between two stimuli (either visual or auditory) that predicted either a food pellet delivery (CS+) or nothing (CS-). In a second phase of the experiment, the authors exposed the rats to a compound training during which both rats (actor and partner) were trained together in the same box divided in two compartments by a transparent barrier. During this phase, the actor rats were exposed to either (1) aCS+ and pCS+ (actor and partner related cues – under this schedule both rats received a pellet); (2) aCS+ and pCS- (only the actor rats received a pellet); or (3) aCS- and pCS- (no pellets for both rats). Finally, the authors presented each cue separately in an extinction probe test showing that once actor rats have fully learned a stimulus-reward association producing reward for themselves, adding a cue that predicted an additional reward delivery to a partner rat unblocked associative learning about that cue. Moreover, by adding an opaque barrier between the two sides of the chamber and preventing visual contact, the authors showed that the pCS+ did not unblock associative learning.I enclose below my comments for the authors.1) The behavioral work, while intriguing, is under-developed. It is not completely clear what is the role of the social partner in this task. The authors should report the behavioral data for this group for two major reasons. Firstly, in a pavlovian task like the one described in this manuscript, there is not a real distinction between actor and partner. Indeed, each actor is the partner of the other rat and vice-versa. Secondly, and probably more importantly, reporting the data of the social partners can be a useful way to show replication of the data (which is a critical feature of a new behavioral procedure). Moreover, the authors should report more data describe the rats' behavior during the task (i.e. is the actor rat watching the partner consuming the pellets? Do both rats enter the food cup at the same time? etc.).2) In line with the above point, it seems that the unblocking effect is driven by the simple presence of a social partner. Indeed, the simple presentation of pCS+ without the visual contact of the social partner, did not unblock associative learning. However, it is hard to disentangle the role of pCS+ from the presence of the social partner only. Therefore, the authors should add a control group in which only the social partner is getting a pellet (aCS- and pCS+). The authors reported some rational to explain why they did not run this experiment; however, this group can show that even without receiving a pellet the pCS+ has a strong effect on the actor rats during the unblocking procedure. This will show a real social transmission effect from the social partner to the actor.3) It is also critical to replicate the finding from the Control group. Given the lower n (8 vs 20 rats in the Experimental group), the variability in this group is bigger than the first experiment and also the magnitude of the effect (for both aCS+ and aCS-) is different relative to the experimental group. This is a major concern considering the reliability of the new model.4) The direct comparisons between experimental and control conditions seems slightly strange. It is hard to direct compare two groups that have been exposed to different behaviors. One group experienced the presence of a social partner and the other did not. I suggest avoiding this statistical comparison, also because the author decided to arbitrarily select 6 trials for this comparison instead of the 10 total trials. Moreover, for the control group the authors reported no interaction between condition type and bin number (Subsection “Control group”), but still reported posthoc comparison. Please clarify.5) The authors should present the data of the auditory vs visual cues in two different lines (and different colors) in the same graph before collapsing them in a single curve. It is important to see that the two modalities are similar.

We will now address the comments of reviewer 2. Firstly, reviewer 2 argues that we should report the partner rat data as there is no real distinction between actor and partner rat. We are convinced that this is not the case in our task setup, as in the main experiment, actor and partner rats intentionally experience different reward contingencies. Our compound conditions in the main experiment include the BR, OR and NR but not the PR. The consequence of this decision makes our task asymmetrical between actors and partners. If a PR condition was included in the current task set, we would assume that the negative aspect of seeing another rat being rewarded, but not oneself (aCS-/pCS+ [PR], see Control Experiment 2), could interact with the unblocking effect observed for BR-associated vicarious value due to mutual reward delivery. Importantly, we do indeed observe that when rewards are disadvantageously unequal for the actor (Partner reward, aCS-, pCS+), unblocking is not observed, possibly due to inequity aversion (Control group 2). We agree to the importance of showing the partner rat data and have therefore included it in figure S2. Partner rats clearly show more responding to the aCS+/pCS- than aCS-/pCS- compound cues, suggesting that, for them, the aCS+ does carry some vicarious value, or drives social facilitation of food cup entries. It is not enough to support formal unblocking as seen for the actors though.

Reviewer 2 furthermore adds that our observed effect could be related to a mere partner presence effect. Here, we would like to again refer to control experiment 2 where we find that observation of another rat being rewarded, while not receiving self-reward does not lead unblocking. This provide evidence against a mere presence interpretation of our social unblocking effect.

Reviewer 2 then addresses the problems associated with replicability due to the small sample size in control experiment 1a (wall condition). We very much agree that is important that this control needs a bigger sample size. We unfortunately had to make a practical choice between enhancing the sample size of experiment 1a or adding the empty cage control. As we think that the empty cage control is crucial for comparison with other studies and broadens the range of control conditions, we decided to include this manipulation. We do want to emphasise that we observe in neither the opaque wall nor the empty-cage a social unblocking effect, and also not when combining both controls as in our main analysis (Control 1A+1B). We hope that this approach will instil confidence that our observed effect is indeed of a vicarious social nature.

Reviewer 2 finally asked for the dissociation between visual and auditory cue learning. We performed additional analyses that indeed show that there is an effect of cue type but that this is limited to the NR condition in the compound phase but not on cue responding in the discrimination learning phase (Figure 1—figure supplement 1).